# A comprehensive study of cognitive control in healthy aging

Sarah De Pue [1] ✉, Hans Stuyck[1], Céline R. Gillebert[1], Eva Dierckx[2,3] & Eva Van den Bussche [1]

## Abstract

**Background** Declines in cognitive control impact older adults' daily life and independence. Cognitive control frameworks distinguish different subcomponents, such as inhibition, updating, shifting, proactive and reactive control. A comprehensive overview of how these subcomponents develop in aging is lacking, with research typically focusing on small samples treated as a homogenous age group, targeting a single subcomponent, or using heterogenous tasks across studies. The aim of the current study was therefore to provide a more comprehensive overview of cognitive control in aging, studying multiple subcomponents in a large sample including different age cohorts.

**Methods** In the current study, young adults ($N = 75$) and three cohorts of older adults ($N = 231$) completed an extensive test battery assessing multiple subcomponents of cognitive control, including response inhibition, interference control, updating, shifting, and reactive and proactive control. Linear mixed models and generalized linear mixed models were used to compare cognitive control performance between the age groups.

**Results** Our results show improved response inhibition and interference control in older compared to young adults. Updating and shifting decrease with age, while proactive control remains preserved in older adults.

**Conclusions** These findings show that changes in cognitive control subcomponents follow different onsets and trajectories, highlighting the importance of including them in cognitive aging research. While some cognitive control functions decline, others seem to be preserved or even improve with age. Longitudinal follow-up studies can help uncover inter- and intra-individual changes in subcomponents with aging.

## Plain language summary

Cognitive control, a crucial set of cognitive functions that enable us to reach our goals, declines with age and impacts older adults' daily life and independence. Living in an aging society, a better understanding of how cognitive control changes with age is needed. However, different aspects of cognitive aging remain poorly understood. Most studies use diverse measurements of cognitive control, include small samples, and only focus on a single cognitive control subcomponent. We administered multiple cognitive control tasks in 231 older adults and 75 young adults. Results showed that not all cognitive control subcomponents declined with age. Some were spared or even improved. This highlights the importance of including multiple cognitive control subcomponents to obtain a comprehensive view on cognitive aging.

One of the biggest challenges of our society is the growing population of older adults. Expectations are that towards 2060, older adults will represent almost a quarter of the population[1,2]. Aging is characterized by declines in cognitive abilities, which can significantly impact daily life functioning and functional independence[3,4], taking a heavy toll on older adults, their family, and society in general. According to Braver and Barch[5], problems in cognitive tasks with aging can be traced back to declines in cognitive control, also referred to as executive functioning. Cognitive control is a broad concept that can be defined as the ability to filter incoming information, inhibit automatic or goal irrelevant information, and maintain and update this information to reach our goals[6]. Without cognitive control, daily life situations such as driving a car, following a conversation without being distracted by background noise[7], planning appointments, or medication intake[8] are impossible, making this a crucial ability for living independently.

Importantly, cognitive control functions mature rather late in life and deteriorate relatively early, linked to the development and deterioration of the prefrontal cortex[9,10]. Hence, older adults are especially prone to experience this negative impact on daily life, making lifespan research into this domain crucial to understand and minimize cognitive declines with older age and improve quality of life[3,11,12].

Developmental changes in cognitive control are extensively studied. Initially, frameworks argued for a general decline in cognitive control with older age[9,13,14]. However, more recent research and meta-analyses often display mixed evidence about the existence of such a general decline (see Dexter et al.[12] for an overview). Rather, whether a decline is observed seems to depend on the specific cognitive control subcomponent under investigation[11,15–18].

One influential theory of cognitive control is the Unity and Diversity framework[19], which distinguishes three subcomponents. *Inhibition* is the

[1]Brain & Cognition, Leuven Brain Institute, KU Leuven, Leuven, Belgium. [2]Personality and Psychopathology, Vrije Universiteit Brussel, Brussels, Belgium. [3]Psychiatric Hospital, Alexianen Zorggroep Tienen, Tienen, Belgium. ✉e-mail: sarah.depue@kuleuven.be

**Article**

**Table 1 | Overview of mixed results, reported by meta-analyses on age effects for inhibition, updating, and shifting**

| | Evidence of declines in the cognitive control subcomponent with age | Evidence of a spared cognitive control subcomponent or improvements with age |
|---|---|---|
| Inhibition | Rey-Mermet and Gade[76]: age-related declines for response inhibition, but not for interference control | Maldonado et al.[38]: no evidence for declines in response inhibition<br>Rey-Mermet et al.[78]: improved interference control in older compared to young adults |
| Updating | Bopp and Verhaeghen[85]: decreased performance in older adults compared to young adults (see also Zuber et al.[22]) | Maldonado et al.[38]: updating the least affected component with age |
| Shifting | Wasylyshyn et al.[86]: age-related deficits depend on the type of switch cost studied (see also Sylvain-Roy et al.[25]) | Verhaeghen and Cerella[77]: no impairments in shifting with age<br>Maldonado et al.[38]: no declines in shifting |

ability to suppress automatic or involuntary responses (i.e., response inhibition) or distractive, irrelevant information (i.e., interference control) to reach one's goal. *Updating* involves holding on to new, relevant information to achieve one's goal, and storing, maintaining, and updating this information in working memory when it is no longer relevant. Finally, *shifting* or switching enables one to flexibly shift between instructions or tasks in order to successfully complete one's goal.

Another framework distinguishing cognitive control subcomponents is the Dual Mechanisms of Control theory[6], defining two modes of cognitive control that differ in the timescale on which they operate. A *proactive control* strategy allows us to actively anticipate and prevent conflict situations, whereas a *reactive control* strategy will be used to respond and overcome conflict only after it has happened. Successfully reaching a goal implies being able to flexibly switch between these strategies and selecting the one that is the most appropriate for a given situation. Goal maintenance is a prerequisite for proactive control, making this strategy more resource-demanding[6]. Proactive control is the optimal strategy in high-conflict situations where conflict can be anticipated, whereas reactive control is the most efficient (resource-preserving) choice when conflict is scarce or unpredictable.

While these two frameworks are not mutually exclusive, both frameworks can inform us about cognitive control in healthy aging. To reliably map cognitive control in aging, these subcomponents of cognitive control should be disentangled. When looking at the lifespan, an inverted-U-shaped curve is found for the development of cognitive control components, showing a decline for inhibition, updating, and shifting in older age[11,18,20,21]. However, meta-analyses report mixed results for each of these components (see Table 1 for an overview). Reported declines seem to go beyond age-related declines in processing speed[18,22,23] (but see Verhaeghen[16]). However, most of the studies assess only one of these cognitive control subcomponents at a time, and due to the different study designs in the literature, it remains unclear which components decline, stabilize, or even improve with age. Still, a few studies have already assessed multiple cognitive control subcomponents in a sample of older adults. Idowu et al.[24] conducted a cross-sectional study in young vs. older adults, assessing inhibition, updating, and shifting. Results showed declines in inhibition, some shifting tasks, and updating with age. The strength of the declines differed depending on the specific function, with inhibition being the most strongly affected. Sylvain-Roy et al.[25] only observed age-related declines for inhibition. In contrast, Zuber et al.[22] showed declines in older adults in all three subcomponents. Finally, Sorel and Pennequin[26] reported that declines in inhibition, shifting, and updating with age differed in onset, pace, and trajectory. Comparing three groups (young adults and two groups of older adults with mean ages of 68 and 79 years), shifting was significantly impaired only in the oldest age group. Inhibition was decreased in both older groups compared to the young adults, and updating declined linearly with age.

Regarding *proactive and reactive control*, the literature has long pointed towards a shift with aging from mainly using *proactive* control to *reactive* control as the preferred strategy[20,27–30]. Studies of Braver et al.[20] and Van Gerven et al.[28] showed context maintenance impairments in older adults aged over 70 years, resulting in reactive control strategies to compensate for this impairment. However, in the past decade, several studies showed that proactive control is not always impaired in older adults[31–34] and might even

be enhanced[35] or that reactive control is impaired with aging[36,37]. Again, declines in proactive and/or reactive control seem to be task-dependent. However, no study so far by our knowledge has directly compared different proactive and reactive control tasks.

In conclusion, cognitive control studies typically only focus on a single component[25,38]. Furthermore, the use of heterogenous cognitive control tasks across studies and the frequent use of neuropsychological tests in older adults (which do not distinguish between subcomponents), lead to mixed findings in the field[23,39]. Moreover, older adults are often treated as a homogenous group, whereas high heterogeneity can be expected in the broad age range from 65 to over 80 years[3,40,41]. In addition, this approach might obscure transitions in cognitive control that only occur for certain age groups (as was the case in the study of Van Gerven et al.[28]). Finally, for those studies that did include multiple cognitive control subcomponents, samples of older adults were often small (e.g., $N = 25$ in Idowu et al.[24], $N = 39$ in Albinet et al.[23], $N = 15$ in Sorel and Pennequin[26]). In other words, a comprehensive view on cognitive control in aging, using multiple tasks to assess multiple subcomponents in more fine-grained older age groups with sufficient power, is missing. Such a comprehensive overview requires an ambitious undertaking, testing a large sample of older adults and using an extensive cognitive control test battery, but it is of crucial importance if we want to understand how cognitive control, as an indispensable skill in daily life, is truly impacted by age.

In this study, we used a cognitive control test battery, assessing differences in response inhibition, interference control, updating, shifting, proactive and reactive control between young adults, older adults between 60 and 69 years, older adults between 70 and 79 years, and older adults aged 80 years or older, similar to the approach of Van Gerven et al.[28]. Based on the literature, we expected a shift from proactive to reactive control around the age of 70 and decreased updating, inhibition, and shifting with age if cognitive control would show a general decline. However, based on the mixed findings in the literature, it could very well be that older adults have intact proactive control and that cognitive control subcomponents decline at different onsets and rates.

This study shows clear age-related declines for updating and shifting. However, proactive control remains preserved with older age, and older adults even outperform young adults on inhibition tasks. No universal decline in cognitive control is observed, highlighting the importance of including multiple subcomponents and a broad age range when studying cognitive control in older adults.

## Methods

### Participants
A group of young adults and three cohorts of older adults were recruited: older adults aged 60–69 (youngest cohort), 70–79 (middle cohort), and 80 years or older (oldest cohort). Young adults were first-year Psychology students from KU Leuven, Belgium, recruited via the Experiment Management System of KU Leuven for course credits. Older adults were recruited across Flanders, Belgium, through family and friends, social media (e.g., forums and sites specifically targeting older adults), and flyers in public places. In addition, existing contacts with participants from a previous study who indicated that they wanted to participate in future research were used to recruit additional participants via telephone or email. For their participation, participants could win a gift certificate via a random draw.

To be eligible for this study, participants had to speak Dutch and have self-reported normal or corrected-to-normal vision and hearing, as visual and auditory information needed to be processed in the study. Moreover, participants could not be colorblind, have no (history of) psychiatric illness or neurodegenerative diseases, no cancer treatment in the last two years, and no drugs or alcohol abuse (all self-reported). Moreover, both the young and older adults had to score 6 or lower on the Geriatric Depression Scale-15 (GDS-15[42]), and older adults had to score 23 or higher on the Montreal Cognitive Assessment (MoCA[43,44]), given the focus on healthy aging. A total of 41 participants were excluded: $N = 2$ were colorblind, $N = 23$ scored below the MoCA cut-off, $N = 14$ scored above the GDS-15 cut-off, $N = 1$ scored both below the MoCA and above the GDS-15 cut-off, and $N = 1$ did not complete the full study. This resulted in a final sample of 306 participants eligible for data analysis: 75 young adults, 80 older adults aged 60–69 years old, 82 older adults aged 70–79 years old, and 69 older adults aged 80 years or older. According to a priori power calculations with MorePower 6.0.4[45], we needed 42 participants per age group to be able to study the interaction in the most complex design (i.e., $3 \times 2$ (within-subject factors Block and Congruency)$\times 4$ (between-subjects factor Age Group) interaction in the Flanker task), assuming a minimal power of 80% to detect small-to-medium effects of magnitude $\eta_p^2 = 0.04$ at the 5% significance level. This sample size criterion was thus more than sufficiently met.

We complied with all relevant ethical regulations. All participants provided written informed consent. The study was approved by the Social and Societal Ethics Committee (SMEC) from KU Leuven (G-2021-3363-R3(AMD)).

## Material

The questionnaires were presented and completed on paper or using Qualtrics[46] (version July 2021–December 2022). Computer tasks were programmed in Python and were administered using PsychoPy[47] (v2020.2.10).

**Montreal Cognitive Assessment (MoCA[43]).** The paper version of the MoCA (version 8.1 in Dutch) was used as a measure of objective cognitive functioning, measuring cognitive decline in executive functioning, visuospatial abilities, attention, concentration, language, short-term memory, and orientation. Total scores range from 0 to 30, with higher scores reflecting higher cognitive functioning. To be eligible for participation, older adults had to achieve a score on the MoCA of 23 or more[44].

**Geriatric Depression Scale-15 (GDS-15[42,48]).** The GDS-15 was assessed to measure depressive symptoms. Participants responded to 15 items with Yes or No. Examples of these items are "Do you feel that your life is empty?" or "Are you afraid that something bad is going to happen to you". Using a score key, each item received a score of 0 or 1, and these scores were then summed, leading to a total score between 0 and 15, with higher scores indicating more depressive symptoms. To exclude the presence of depressive symptoms, we only included participants with a score of 6 or lower on this scale.

**Cognitive control test battery.** The test battery assessing inhibition, updating, shifting, and proactive and reactive control is available on the OSF page of this study[49], https://doi.org/10.17605/OSF.IO/VBH58. All tasks were programmed using visual degrees as a unit, except for the Plus–Minus task, which had pixels as a unit. At the beginning of each task, participants had to fill in their participant number, gender, age, and handedness. In addition, the screen width of the laptop on which the tasks were administered was entered to make sure that stimulus sizes were identical for all participants, independent of the laptop used for data collection.

**Go/No-Go task.** In the Go/No-Go task[50], measuring response inhibition, colored squares (either blue or yellow) were presented on the screen. Participants were instructed to press "J" as fast as possible if the square was blue (=Go trial) and to withhold a key press if the square was yellow (=No-Go

trial). This rule was counterbalanced across participants. Eighty percent of the trials were Go trials, the other 20% were No-Go trials. Participants completed one practice block of 20 trials and four main blocks of 50 trials. Between each block, a break was provided. Feedback was provided for 1000 ms after each practice trial ("wrong" in red, "correct" in green, or "too slow" in black) and after incorrect main trials. On each trial, a fixation cross was first presented for 500 ms (size = 1.125), after which a blue or yellow square appeared on the screen (width = 4, height = 4). The response deadline was 500 ms. The dependent variables were the reaction times (RTs) and errors (i.e., 0 = correct, 1 = incorrect) at the trial level. Two types of errors were measured: omission errors (percentage of errors on Go trials), where the participant should have pressed the key but did not, and commission errors (percentage of errors on No-Go trials), where the participant pressed the key but should have inhibited this response. Commission errors are often used as the main index of response inhibition in this task[51].

**Flanker task.** In the Flanker task[52], measuring interference control and proactive and reactive control, participants received a string of seven numbers on each trial and had to indicate as fast and accurate as possible whether the central number, i.e. the target, was 1 or 2 (by pressing "F" with the left hand) or 3 or 4 (by pressing "J" with the right hand). The target number was flanked by distractor numbers (three on each side). Trials were either congruent, when the target and distractors triggered the same response (e.g., 1112111 or 2222222), or incongruent, when the target and distractors triggered an opposite response (e.g., 4442444 or 1113111). Thus, on incongruent trials, conflict was present and cognitive control was needed to overcome it. Incongruent trials are typically responded to slower and less accurate than congruent trials, creating a flanker or congruency effect[53], the index of interference control. Crucially, the proportion of incongruent trials was manipulated in separate blocks of trials. Participants received three blocks: a block with 87% congruent trials (87C block), a block with 67% congruent trials (67C block), and a block with 47% congruent trials (47C block). A low proportion of incongruent trials triggers the use of reactive control, as conflict is rare and cannot be anticipated. This leads to efficient responses to the frequent congruent trials, but slow and less accurate responses on the rare incongruent trials, leading to a large congruency effect in this block. A high proportion of incongruent trials allows participants to anticipate the frequent conflict, making proactive control the optimal strategy, which is reflected in a smaller congruency effect in this block[54]. The decreased congruency effects in blocks with high as compared to low proportions of conflict is referred to as the Proportion Congruency Effect or PCE[17]. The order of blocks was counterbalanced across participants. Participants first completed 15 practice trials, followed by 3 blocks of 120 trials with a break in between. On each trial, a fixation cross was presented for 1000 ms (size = 1.125), followed by the flanker stimulus that stayed on the screen until the participant responded (size = 1.125). Feedback was provided after practice trials ("wrong" in red or "correct" in green). The dependent variables were the RTs and errors at the trial level, which enabled us to analyze the mean RTs and the percentage of errors across the different conditions. The 47C block was used to study interference control, whereas all blocks were included in the analyses of proactive and reactive control.

**N-Back task.** In the N-Back task[55], measuring updating, gray-scale pictures of objects and animals were presented serially. Participants had to decide for each stimulus whether the presented picture was identical to the picture that was presented second-to-last (i.e., 2-back task), by pressing "J" as quickly as possible if that was the case (i.e., target trials). All used pictures were validated in a previous study of Rossion and Pourtois[56]. Participants received one practice block of 50 trials and two main blocks of 50 trials with a break in between, both with a 30% rate of target trials (i.e., 15 target trials and 35 non-target trials per block). Per trial, images (size 6.24 × 4.38) were presented until response or until the response deadline of 1500 ms was exceeded. In between images, i.e., the inter-stimulus interval, a blank screen was presented for 500 ms. Feedback was provided after each practice trial and after incorrect main trials for 1000 ms ("wrong" in red, "correct" in green,

size = 1.125). The dependent variables were RTs and errors at the trial level. This enabled us to analyze mean RTs on the target trials, the percentage of errors on target trials (when the participant should have pressed the key but did not, i.e., omission errors or missed targets), and the percentage of errors on non-target trials (when the participant pressed the key but should not have, i.e., commission errors or false alarms). Omission errors or missed targets are often used as the main index of updating in this task[57].

Plus–Minus task. In the Plus–Minus task[58,59], measuring shifting, a list of random two-digit numbers was presented. In a first block, participants had to add 3 to each two-digit number (ranging from 13 to 96), in a second block, they had to subtract 3 from each number, and in a last block, they had to alternate between adding 3 and subtracting 3. Participants received a list of 6 numbers as a practice list for each block. In each main block, participants completed a list of 30 numbers (size = 40), presented on a single screen (in three columns of 10 numbers). Participants were asked to type the answer next to each number. Feedback was presented after completing a list. For each list, the completion time of the full list and the total number of errors per list were measured. The dependent variables were the required time to complete a list (in seconds) and the total number of errors for each list, which enabled us to compare these outcomes between the non-switch and the switch blocks at the participant level. A switch cost was calculated as the difference between the completion time or errors on the switch block and the average completion time or errors on the no switch blocks, i.e., the global switch cost as an index of shifting. Due to the design of the Plus–Minus task, with only total time to complete a list and total accuracy per list, we were not able to compute a local switch cost.

AX-CPT. In the AX-Continuous Performance Task (AX-CPT[60]), measuring proactive and reactive control, a series of letters was presented on the screen: First, a cue, either A or B (B represents any letter except A), followed by a probe, which is either X or Y (Y represents any letter except X). Participants needed to respond as quickly and accurately as possible to the probe with a "J" key press for target trials (i.e., trials where A cue is followed by X probe) and a "K" key press to all other, non-target trials. Response mapping was counterbalanced. Four types of trials were created: (1) AX trials (70% of the trials), where a probe X is preceded by a valid cue A. As AX trials are presented with a high frequency, a high expectancy for an X probe is created when receiving an A cue. (2) BX trials (10%), where a valid probe X is preceded by an invalid cue B. (3) AY trials (10%), where a valid A cue is followed by an invalid Y probe. (4) BY-trials (10%), which serve as a control condition. The pattern of RTs and error rates on AY and BX trials can provide insight into the used cognitive control strategy. Adopting a proactive control strategy, actively maintaining the valid A cue and thus preparing for a target response will lead to worse performance when encountering an invalid target, i.e., AY trials. Performance on BX trials will, however, be good using this strategy, as the invalid B cue triggers planning of the correct non-target response. The opposite pattern can be expected when adopting reactive control, where participants will react slower and less accurate on BX trials, where valid X probes bias the preceding invalid cue towards an erroneous target response, but show good performance on AY trials, where invalid Y probes bias the preceding valid cue towards the correct non-target response[61]. Participants completed 20 practice trials and two blocks of 100 main trials with a break in between. On each trial, a fixation cross (size = 1.125) was first presented for 1000 ms, followed by a cue (size = 1.125) for 1000 ms, an inter-stimulus-interval of 2000 ms, and a probe (size = 1.125) until the participant responded or until a response deadline of 2500 ms was exceeded. Feedback was provided after practice trials ("wrong" in red and "correct" in green). The dependent variables were RTs and errors at the trial level, enabling us to analyze the mean RT and percentage of errors within each condition. Additionally, the balance in performance between AY trials and BX trials was captured with the Proactive Behavioral Index (PBI), calculated as $(AY-BX)/(AY+BX)$. A positive index indicates a preference for proactive control, whereas a negative index reflects a preference for reactive control[27,62]. The PBI can inform us about a preference for either proactive or reactive control, whereas performance on AY and BX trials can inform us about proactive/reactive control performance in young vs. older adults.

General and demographic questions. Next to age and gender, participants were asked to list diseases that could affect their cognitive abilities. In addition, socio-economic status was reported, with questions about education, profession, and income. Furthermore, computer use was administered. Finally, participants indicated with Yes or No if they had been more forgetful in the past year to the extent that it has significantly affected their daily life, and gave an example of daily life, to measure subjective cognitive functioning[63]. These questions were used to characterize the sample of this study (see Supplementary Tables 1 and 2).

## Procedure
Participants were tested at home, keeping in mind the COVID-19 regulations that were in place during the data collection of this study (June 2021 until January 2023). An experimenter was present during data collection. The MoCA and GDS-15 were administered first, on paper, to check if participants met the inclusion criteria. If older adults scored lower on the MoCA cut-off score and/or higher on the GDS-15 cut-off score, they were excluded from this study. In this case, they still executed the Reaction Time task, instead of the full test battery, to mitigate negative feelings of not being included in the study. The older adults meeting the inclusion criteria completed the full test battery. The order of the tasks was counterbalanced across participants according to a Latin square. Next to the tasks discussed above, participants also completed a cognitive-motor-dual task and a basic Reaction Time task. The experimenter first explained the task and monitored practice trials, whereafter participants completed the task themselves. An external keyboard and mouse were used in addition to the laptop to administer the tasks. After each task completion, participants reported their subjective experience using the Performance and Effort scale of the NASA Task Load Index (NASA-TLX[64,65]). After the cognitive control test battery, participants filled in the additional questionnaires in Qualtrics (i.e., the general and demographic questions and the Lubben Social Network Scale-6, LSNS-6[66], to assess social support). At the end of the session, the forward and backward digit span of the Wechsler Adult Intelligence Scale[67,68] (WAIS-IV) were assessed as a measure of objective cognitive functioning. The testing session took between two to three hours, with breaks provided in between when needed. The cognitive-motor-dual task, basic Reaction Time task, NASA-TLX scales, LSNS-6, and WAIS-IV digit span go beyond the scope of this study and won't be discussed here. We refer other researchers to the OSF page[49] (https://doi.org/10.17605/OSF.IO/VBH58) to further explore these data, only excluding those data that are still embargoed as they are a part of a different study (i.e., cognitive-motor-dual task).

## Statistics and reproducibility
First, for RT analyses, practice trials and incorrect trials were removed. Next, trials slower than 2.5 SD above a participant's individual mean RT were removed. This resulted in the removal of 6.67% incorrect and 1.10% too slow trials for the Go/No-Go, 2.54% incorrect and 2.34% too slow trials for the Flanker, 22.26% incorrect and 1.39% too slow trials for the N-Back, and 5.72% incorrect and 2.56% too slow trials for the AX-CPT across all Age Groups. For error rate (ERR) analyses, practice trials were removed from the data. Finally, to ensure data reliability, participants were excluded from the analysis of a specific cognitive control subcomponent if they had fewer than 25% valid data points relative to the total number of trials in each condition. This led to the exclusion of no participants for inhibition, $N = 8$ for updating, $N = 4$ for shifting, and $N = 13$ for proactive/reactive control. When adopting a stricter cut-off of 50%, results remained highly similar. As this significantly reduced sample sizes, especially for the oldest age group, the 25% criterion was retained. In the following explanation of the statistical analyses, all significant main and interaction effects identified through analysis of variance (ANOVA), linear mixed models (LMM), and generalized linear mixed models (GLMM) were followed by post-hoc tests with Tukey correction for

## Table 2 | Participant characteristics for each Age Group

| Age group | N | Mean age (SD) | Age range | Gender (% female) | MoCA | GDS-15 | Education (number of years) | Education range |
|---|---|---|---|---|---|---|---|---|
| Young adults | 75 | 18 (1.32) | 18–27 | 84 | – | 2.67 (1.90) | 13.03 (1.51) | 12–18 |
| 60–69 years old | 80 | 65 (2.77) | 60–69 | 55 | 26.94 (2.28) | 1.55 (1.47) | 14.38 (2.72) | 6–19 |
| 70–79 years old | 82 | 74 (2.63) | 70–79 | 59 | 25.88 (1.75) | 1.85 (1.39) | 13.22 (3.58) | 6–24 |
| 80+ years old | 69 | 84 (3.47) | 80–95 | 57 | 25.65 (2.03) | 1.83 (1.30) | 13.10 (2.69) | 8–20 |

*SD* standard deviation, *MoCA* Montreal Cognitive Assessment (range 0–30), *GDS-15* Geriatric Depression Scale-15 total score (range 0–15).

multiple comparisons. Cohen's *d* effect sizes were also provided for these pairwise comparisons. Based on Cohen's[69] effect sizes of 0.20, 0.50, and 0.80 were considered as small, medium, or strong, respectively.

First, participant characteristics per Age Group were summarized, and Age Groups were compared on gender and MoCA scores to characterize each Age Group, using one-way ANOVAs.

Second, LMMs and GLMMs were used to analyze the difference between Age Groups in inhibition, updating, shifting, and proactive/reactive control. For RTs, LMMs were applied to the continuous data using a Gaussian error distribution. When assumptions were violated (i.e., for the Go/No-Go and Flanker tasks), Box-Cox transformations on the RTs were applied (for a review, see Atkinson et al.[70]). For the analyses of the errors, GLMMs were applied to the binary data (0 = correct and 1 = incorrect) using a binomial error distribution. The models included main and interaction effects for all tasks, using sum coding to set the contrasts for the main effects. The significance of the main and interaction effects was assessed with a Type III ANOVA with the Satterthwaite approximation method for the LMMs and the Wald test for the GLMMs. The random structure for each (G)LMM was built stepwise. All models included random intercepts for participants, and random slopes for task conditions were added stepwise[71,72]. To obtain the model with the best-fitting random structure, models were compared with likelihood ratio tests. Akaike information criterion (AIC) was used to select the best model (i.e., with the lowest AIC; see Supplementary Table 3). The final random structure for each task is described below. An example equation of the LMM for the reaction times for one of the tasks is provided in the Supplementary Methods, which can be extended to other models in the manuscript as well.

Third, in the Plus–Minus task, error rates are count data (i.e., the total number of errors in a specific block), and thus, a GLMM with Poisson error distribution was used to account for this.

*Inhibition* was studied using the Go/No-Go task (RTs, omission errors, and commission errors) and the 47C block of the Flanker task (RTs and errors). For the Go/No-Go task, for both RTs and errors, the best-fitting models consisted of a single fixed effect for the Age Group (4 levels: young adults, 60–69, 70–79, and 80+) and random intercepts for participants. For the Flanker task, fixed effects in these models were Congruency (2 levels: congruent vs. incongruent), Age Group (4 levels: young adults, 60–69, 70–79, and 80+), and their interaction, with random intercepts for participants and random slopes for Congruency. The error model only included random intercepts for participants. Significant interaction effects between Age Group and Congruency were further explored by calculating the congruency effect (CE; incongruent–congruent) from predicted means, followed by a one-way ANOVA to compare this effect across Age Groups.

To study *updating*, the N-Back task (RTs, errors on target and nontarget trials) was analyzed. These models included a single fixed effect for the Age Group (4 levels: young adults, 60–69, 70–79, and 80+) with random intercepts for participants.

For *shifting*, the time to complete the list (in seconds) and the total number of errors (ranging from 0 to 30) on the Plus–Minus task were analyzed. For all models, Age Group (4 levels: young adults, 60–69, 70–79, and 80+), Switch (switch vs. no switch block), and their interaction were included as fixed effects, with random intercepts for participants and random slopes for Switch. Significant interactions between Switch and Group were further studied by calculating the switch cost (Time needed or errors on switch block—average Time needed or errors on no switch blocks) from

predicted means, followed by a one-way ANOVA to compare this difference across Age Groups.

Finally, to study *proactive and reactive control*, the AX-CPT and Flanker with proportion congruency manipulation were analyzed. For the AX-CPT, models for both RTs and errors included Age Group (4 levels: young adults, 60–69, 70–79, and 80+), Trial Type (4 levels: AX, AY, BX, BY), and their interactions as fixed effects, with random intercept for participants and random slopes for Trial Type. Additionally, a proactive behavioral index (PBI) was calculated, with the formula $(AY-BX)/(AY+BX)$ for both RTs and errors, and subsequently compared across Age Groups using a one-way ANOVA. For the Flanker task, models for both RTs and errors included Block (3 levels: 87%, 67%, 47% congruency), Congruency (2 levels: congruent, incongruent), Age Group (4 levels: young adults, 60–69, 70–79, and 80+), and their interaction as fixed effects. The random structure for RTs included random intercepts for participants and slopes for Block and Congruency, while the errors' model included random intercepts for participants and slopes for Congruency. The main result section reports only the significant highest-order interaction: a three-way interaction between Block, Congruency, and Age Group (significant lower-order two-way interactions are discussed in the supplementary materials). This three-way interaction was further explored by calculating the Proportion Congruency Effect (PCE; CE 87C block–CE 47C block) from predicted means, followed by a one-way ANOVA to compare this difference score across Age Groups.

All analyses were performed in R[73] (v4.2.1) and RStudio[74] (v2024.4.2.764), with the lme4 package[72] (v1.1.31) and the emmeans package[75] for post-hoc tests (v1.8.5).

## Results

### Participant characteristics

Table 2 (for the whole sample) and Table 3 (for the analyzed sample per cognitive control subcomponent) provide a detailed overview of age, gender, and mean scores on the questionnaires of interest per group. Age Groups significantly differed in gender distribution ($X^2(3) = 18.39$, $p < 0.001$), with more female participants in the young age group compared to the older groups. Older adult groups significantly differed in MoCA scores ($F(2,228) = 8.83$, $p < 0.001$). Older adults aged 60-69 had significantly higher MoCA scores than the two oldest age groups ($p \leq 0.0029$). Moreover, differences in scores on the GDS-15 ($F(3,302) = 7.51$, $p < 0.001$) and in education ($F(3,301) = 4.14$, $p = 0.0068$) were observed between groups. Young adults had significantly higher GDS-15 scores than the older adults ($p \leq .0067$). When excluding participants who scored higher than 4 on the GDS-15, no significant differences in depressive symptoms were detected anymore between groups ($p = 0.065$). However, as the pattern of results remained highly similar and the sample size was significantly decreased because of this additional exclusion, we opted to include all participants in the study. In addition, older adults aged 60–69 reported having a higher education compared to the other age groups ($p \leq 0.042$). Note that comparisons with the young adults group are not very meaningful, as this group's education was still ongoing. Supplementary Tables 1 and 2 provide additional descriptive demographic and questionnaire information per group. Based on self-reported diseases that could affect cognitive functioning, some participants could be subject to exclusion from analyses (due to stroke or chronic depression). However, as results remained highly

**Table 3 | Participant characteristics for each Age Group for each cognitive control component**

| Age group | Inhibition (Go/No-Go, Flanker) | | | Updating (N-Back) | | | Shifting (Plus-Minus) | | | Proactive vs. reactive control (Flanker, AX-CPT) | | |
|---|---|---|---|---|---|---|---|---|---|---|---|---|
| | N | Gender (% female) | Mean age (SD) | N | Gender (% female) | Mean age (SD) | N | Gender (% female) | Mean age (SD) | N | Gender (% female) | Mean age (SD) |
| Young adults | 75 | 84 | 18 (1.32) | 75 | 84 | 18 (1.32) | 74 | 84 | 18 (1.32) | 72 | 85 | 18 (1.34) |
| 60-69 years old | 80 | 55 | 65 (2.77) | 77 | 56 | 65 (2.80) | 80 | 55 | 65 (2.77) | 79 | 56 | 65 (2.77) |
| 70-79 years old | 82 | 59 | 74 (2.63) | 80 | 60 | 74 (2.66) | 80 | 59 | 74 (2.65) | 78 | 56 | 74 (2.64) |
| 80+ years old | 69 | 57 | 84 (3.47) | 64 | 58 | 84 (3.50) | 67 | 57 | 84 (3.50) | 63 | 59 | 84 (3.17) |

SD standard deviation.

similar when excluding these participants, we opted to include them in the study.

An overview of the results for age differences in inhibition, updating, shifting and proactive/reactive control is shown in Figs. 1–4 and in Table 4.

**Age differences in inhibition**

**Go/No-Go task.** The LMM for Box-Cox transformed reaction times (RTs) showed a main effect of Age Group ($F(3,301.26) = 178.88$, $p < .001$). Post-hoc comparisons showed that with age participants reacted significantly slower. Young adults ($M = 277$ ms) were faster than the 60–69 ($M = 351$ ms; $t(301) = -16.42$, $p < 0.001$, $d = -1.47$), 70–79 ($M = 362$ ms; $t(301) = -18.75$, $p < 0.001$, $d = -1.67$) and 80+ groups ($M = 375$ ms; $t(302) = -20.67$, $p < 0.001$, $d = -1.92$). The 60-69 ($t(302) = -4.94$, $p < 0.001$, $d = -0.45$) and 70–79 age group ($t(302) = -2.78$, $p = 0.030$, $d = -0.25$) were faster than the 80+ group. The difference in RTs between older adults of 60–69 and 70–79 years old was not significant ($p = 0.11$).

The GLMM for omission errors showed a main effect of Age Group ($X^2(3) = 162.58$, $p < 0.001$). Post-hoc comparisons showed that with age, significantly more omission errors were made. Young adults ($M = 1.09\%$) made fewer omission errors than the 60–69 ($M = 3.06\%$; $Z = -5.66$, $p < 0.001$, $d = -1.05$), 70–79 ($M = 4.92\%$; $Z = -8.48$, $p < 0.001$, $d = -1.54$), and 80+ groups ($M = 10.05\%$; $Z = -12.41$, $p < 0.001$, $d = -2.31$). Older adults between 60 and 69 made fewer errors than the 70–79 ($Z = -2.95$, $p = 0.017$, $d = -0.50$) and 80+ group ($Z = -7.34$, $p < 0.001$, $d = -1.27$), whereas older adults between 70 and 79 years old made fewer omission errors than older adults over the age of 80 ($Z = -4.56$, $p < 0.001$, $d = -0.77$).

The GLMM for commission errors showed a main effect of Age Group ($X^2(3) = 14.92$, $p = 0.0019$). Post-hoc comparisons showed that young adults ($M = 11.58\%$) made more commission errors than the 70–79 group ($M = 6.57\%$; $Z = 3.70$, $p = 0.0012$, $d = 0.62$). There were no significant differences between the other Age Groups ($M_{60–69} = 7.89\%$, $M_{80+} = 9.37\%$, all $p \geq 0.053$).

**Flanker task.** The LMM for Box-Cox transformed RTs showed a significant main effect of Congruency ($F(1,299.15) = 160.89$, $p < 0.001$), showing that participants reacted slower on incongruent ($M = 647$ ms) compared to congruent trials ($M = 627$ ms), i.e., a congruency effect. Post-hoc comparisons for the main effect of Age Group ($F(3,301.81) = 82.60$, $p < 0.001$) showed that with age, participants reacted significantly slower. Young adults ($M = 512$ ms) were faster than the 60-69 ($M = 661$ ms; $t(302) = -11.00$, $p < 0.001$, $d = -1.41$), 70–79 ($M = 673$ ms; $t(302) = -11.75$, $p < 0.001$, $d = -1.50$) and 80+ groups ($M = 736$ ms; $t(302) = -14.66$, $p < 0.001$, $d = -1.95$). The 60–69 ($t(302) = -4.13$, $p < 0.001$, $d = -0.54$) and 70–79 age group ($t(302) = -3.48$, $p = 0.0032$, $d = -0.45$) were faster than the 80+ group. The difference in RTs between older adults of 60–69 and 70–79 years old was not significant ($p = 0.90$). Finally, a significant interaction effect between Congruency and Age Group ($F(3,299.01) = 6.53$, $p < 0.001$) was found. To further explore this interaction, congruency effects (CE) were calculated and compared between Age Groups. The CE was larger in young adults ($M = 21.70$ ms) compared to older adults between 60 and 79 years old ($M_{60–69} = 11.80$ ms, $M_{70–79} = 16.50$ ms, $p \leq 0.0017$, $d \geq 0.75$). Moreover, the CE was smaller in the 60–69 ($M = 11.80$ ms) compared to the 70–79 ($M = 16.50$, $p = 0.0044$, $d = -0.48$) and 80+ Age Groups ($M = 19.30$ ms, $p < 0.001$, $d = -0.72$). The CE did not differ significantly between the young adults and the 80+ cohort and between the two oldest age cohorts ($p \geq 0.20$).

The GLMM for errors showed a significant main effect of Congruency ($X^2(1) = 6.48$, $p = 0.011$), with participants making more errors on incongruent ($M = 1.89\%$) than congruent trials ($M = 1.57\%$), i.e., the congruency effect. Post-hoc comparisons for the main effect of Age Group ($X^2(3) = 57.16$, $p < 0.001$) showed that young adults ($M = 4.07\%$) made more errors on the Flanker task than the 60–69 ($M = 0.91\%$; $Z = 8.83$, $p < 0.001$, $d = 1.53$), 70–79 ($M = 1.46\%$; $Z = 6.59$, $p < 0.001$, $d = 1.05$) and

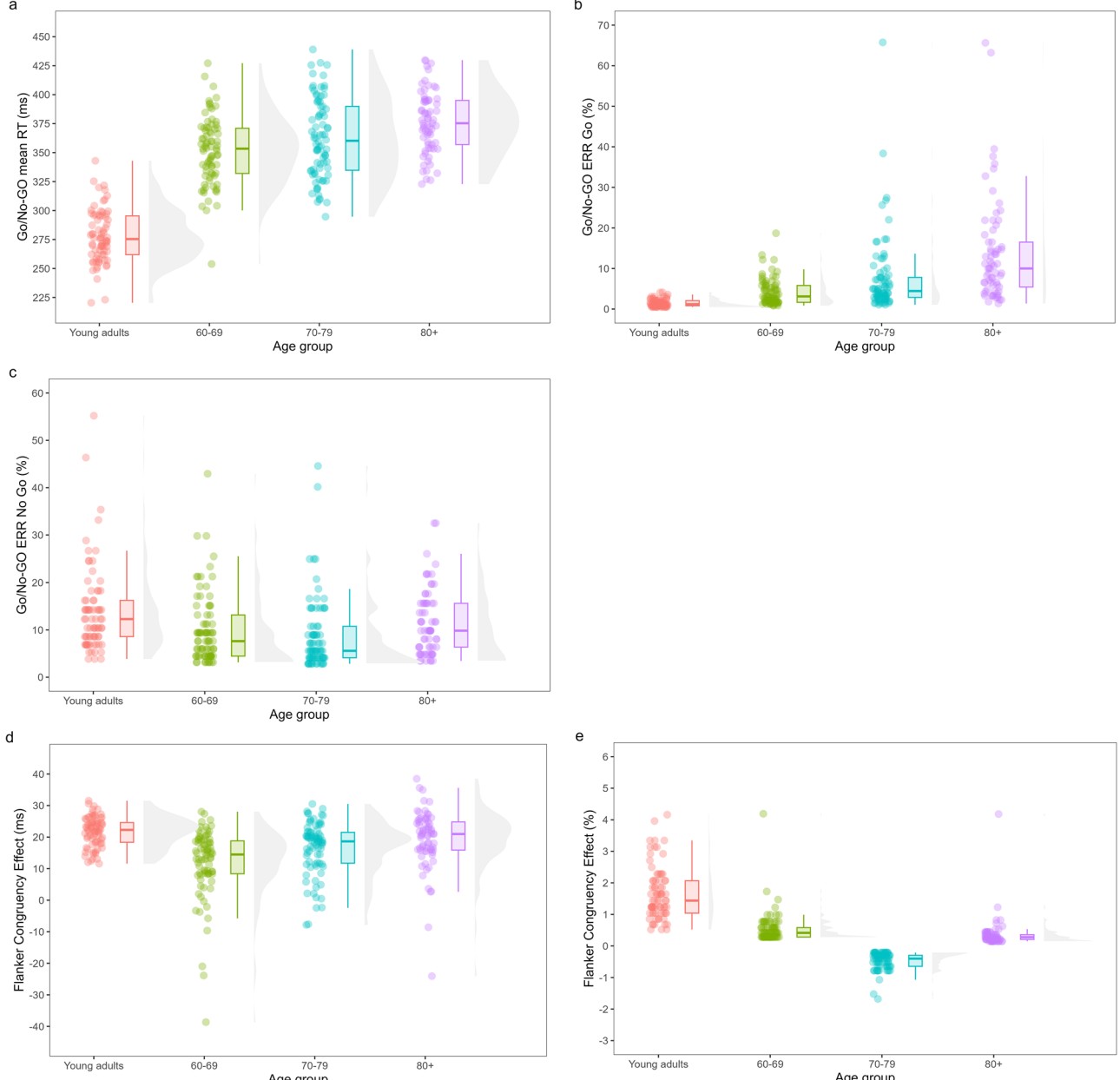

**Fig. 1 | Age differences in response inhibition measured by the Go/No-Go and interference control measured by the 47C block of the Flanker task. a–c** shows the results of the Go/No-Go task. **a** shows reaction times on the Go trials, **b** shows omission errors or errors on the Go trials and **c** shows the commission errors or errors on the No-Go trials. **d** and **e** shows the results of the 47C block of the Flanker task. **d** shows the congruency effect calculated for reaction times and **e** shows the congruency effect calculated for error rates.

80+ groups ($M = 1.61\%$; $Z = 5.70$, $p < 0.001$, $d = 0.95$). Older adults between 60 and 69 made fewer errors than the 70–79 ($Z = -2.64$, $p = 0.042$, $d = -0.48$) and 80+ group ($Z = -3.09$, $p = 0.011$, $d = -0.58$). There were no significant differences in errors between older adults aged 70–79 and 80 years or older ($p = 0.94$). The significant interaction effect between Congruency and Age Group ($X^2(3) = 14.58$, $p = 0.0022$) was further explored by comparing the congruency effect between Age Groups ($F(3,302) = 190.9$, $p < 0.001$). More specifically, the CE was larger in young adults ($M = 1.68\%$) compared to all older adult cohorts ($M_{60-69} = 0.56\%$, $M_{70-79} = -0.46\%$, $M_{80+} = 0.37\%$, $p \leq 0.020$, $d \geq 0.31$). In addition, the CE of the 70–79 group was smaller compared to that of the other older age cohorts (all $p < 0.001$, $d \geq 1.64$). There were no significant differences in CE between the 60–69 and 80+ group ($p = 0.19$).

To sum up, although older adults showed decreased general performance on both tasks (i.e., slower RTs and more omission errors on the Go/

No-Go), their inhibitory control was superior compared to young adults (i.e., smaller congruency effects and fewer commission errors). In addition, for RTs on the Flanker task, interference control was the most efficient in the 60-69 group compared to the other older adult groups. For error rates, however, the 70–79 group showed the most efficient interference control within the older adult groups.

### Age differences in updating
**N-Back task**. The LMM for RTs on correct target trials or Hits showed a main effect of Age Group ($F(3,284.63) = 55.49$, $p < 0.001$). With age, participants slowed down on the N-Back task. Young adults ($M = 604$ ms) were faster than the 60–69 ($M = 732$ ms; $t(277) = -6.88$, $p < 0.001$, $d = -0.60$), 70–79 ($M = 800$ ms; $t(281) = -10.59$, $p < 0.001$, $d = -0.92$) and 80+ groups ($M = 830$ ms; $t(284) = -11.52$, $p < 0.001$, $d = -1.06$). Older adults between 60 and 69 years old were faster than the

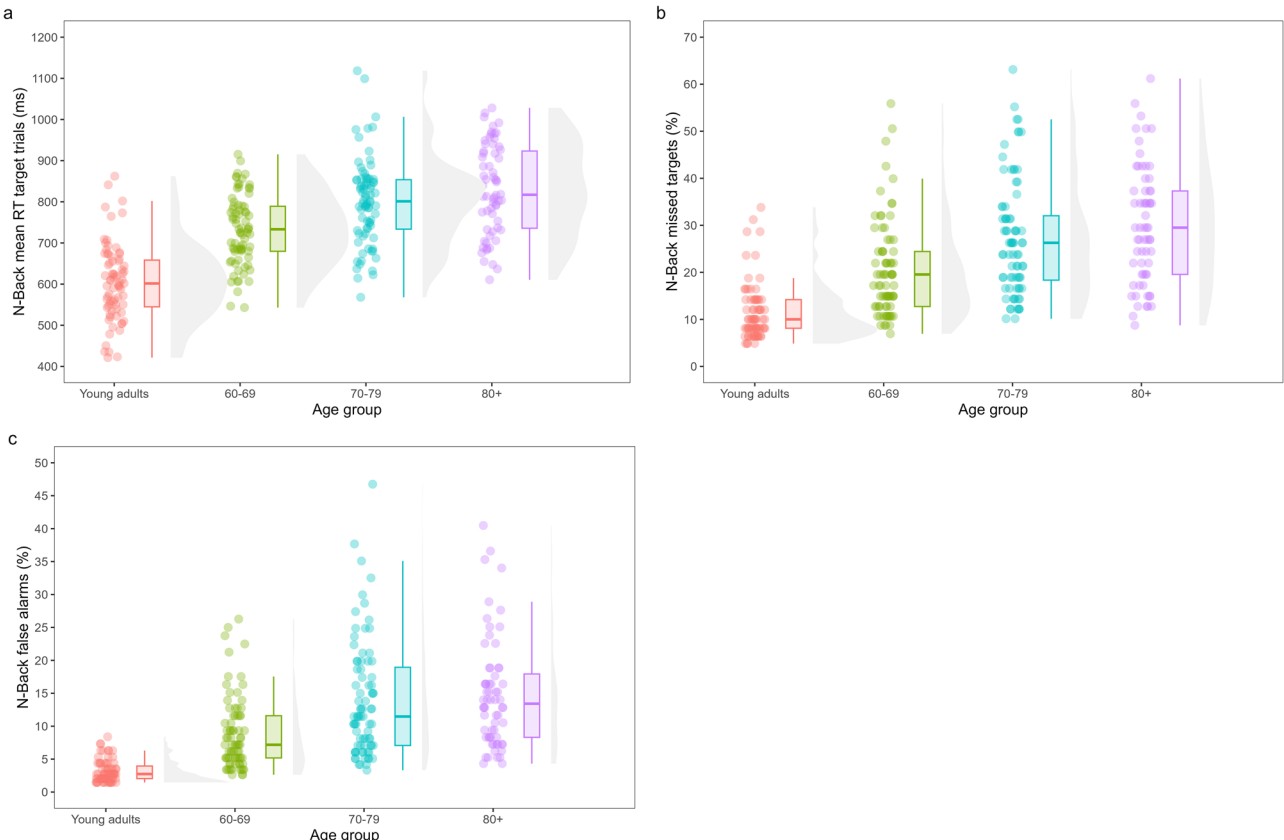

**Fig. 2 | Age differences in updating measured by the N-Back task. a** Shows reaction times on the target trials, **b** shows the missed targets or errors on target trials, and **c** shows the false alarms or errors on the non-target trials.

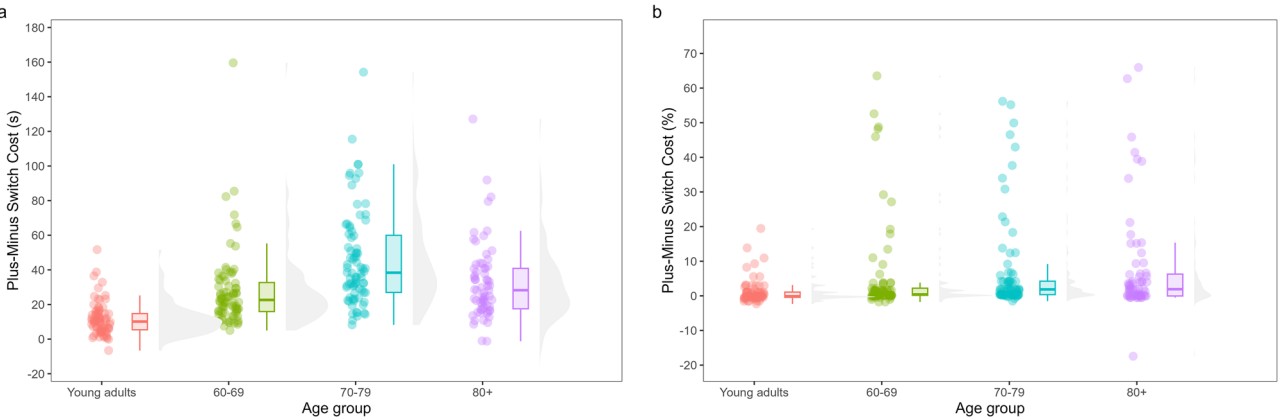

**Fig. 3 | Age differences in shifting measured by the Plus–Minus task. a** Shows switch costs calculated for reaction times and **b** shows switch costs calculated for error rates.

70–79 ($t(286) = -3.67$, $p = 0.0016$, $d = -0.32$) and 80+ age group ($t(289) = -5.01$, $p < 0.001$, $d = -0.46$). The difference in RTs between older adults of 70–79 and 80+ years old was not significant ($p = 0.40$).

The GLMM for missed targets showed a main effect of Age Group ($X^2(3) = 81.50$, $p < 0.001$). Post-hoc comparisons showed that with age significantly more targets were missed. Young adults ($M = 10.00\%$) made fewer errors than the 60–69 ($M = 18.40\%$; $Z = -4.77$, $p < 0.001$, $d = -0.71$), 70–79 ($M = 25.10\%$; $Z = -7.57$, $p < 0.001$, $d = -1.10$) and 80+ groups ($M = 27.90\%$; $Z = -8.16$, $p < 0.001$, $d = -1.24$). Older adults between 60 and 69 made fewer errors than the 70–79 ($Z = -2.85$, $p = 0.023$, $d = -0.39$) and 80+ group ($Z = -3.69$, $p = 0.0013$, $d = -0.54$). The percentage of missed targets did not differ significantly between the two oldest age cohorts ($p = 0.75$).

The GLMM for false alarms showed a main effect of Age Group ($X^2(3) = 138.39$, $p < 0.001$). Young adults ($M = 2.61\%$) made fewer errors on non-target trials than the 60–69 ($M = 7.28\%$; $Z = -6.81$, $p < 0.001$, $d = -1.08$), 70–79 ($M = 11.65\%$; $Z = -10.34$, $p < 0.001$, $d = -1.59$), and 80+ age groups ($M = 12.71\%$; $Z = -10.52$, $p < 0.001$, $d = -1.69$). Moreover, the 60–69 group showed fewer false alarms than the 70–79 ($Z = -3.71$, $p = 0.0012$, $d = -0.52$) and 80+ group ($Z = -4.20$, $p < 0.001$, $d = -0.62$). There were no significant differences between the two oldest age groups ($p = 0.90$).

To summarize, performance on the N-Back task clearly showed age-related declines in updating, reflected in higher errors with age on target and non-target trials (i.e., more misses and false alarms) and slower RTs overall with age. This age-related decline seemed to stabilize from 70 to 79 years, as the two oldest age groups did not differ significantly.

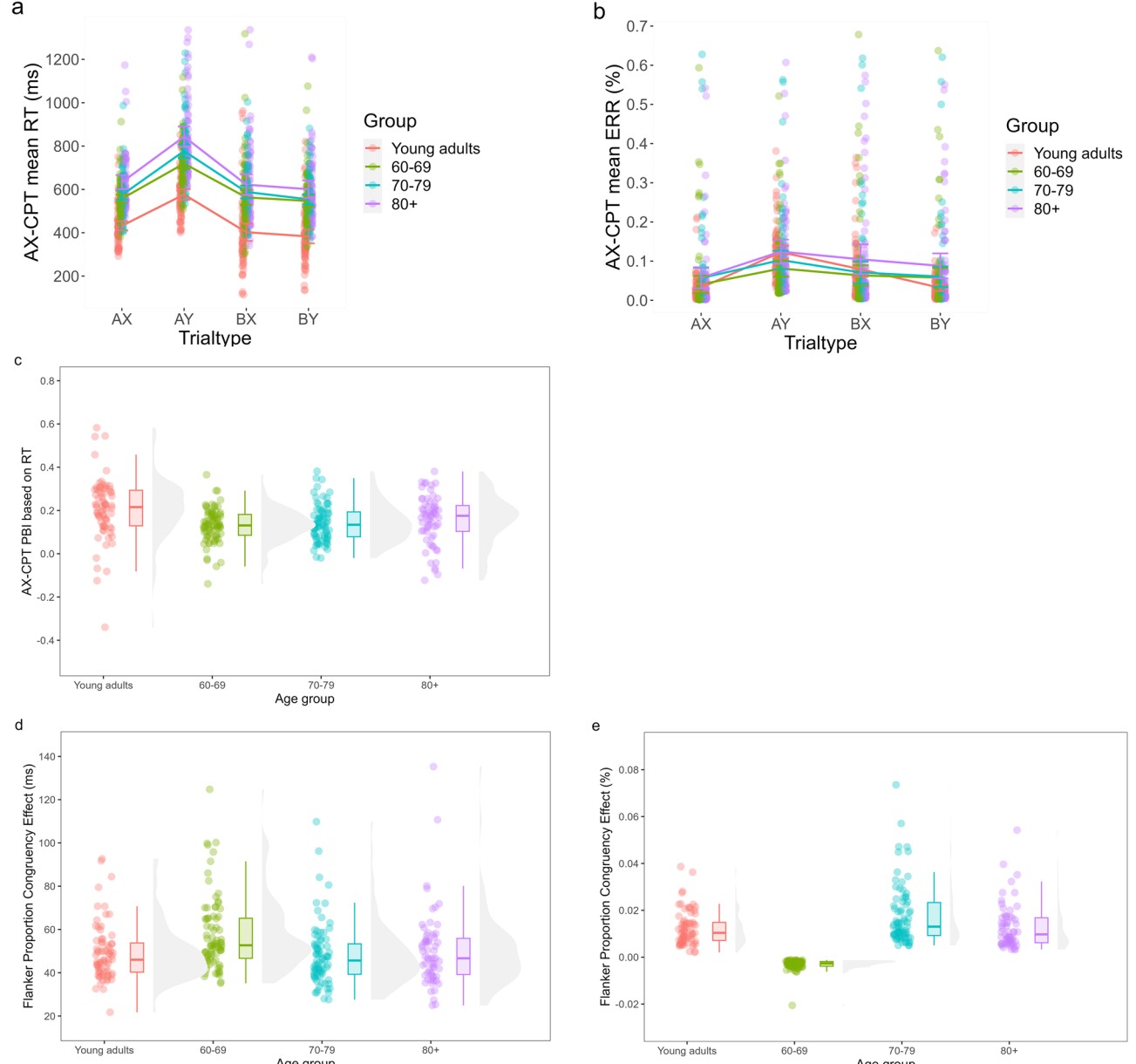

**Fig. 4 | Age differences in proactive and reactive control measured by the AX-CPT and Flanker task with proportion congruency manipulation. a–c** Shows the results of the AX-CPT. **a** Shows reaction times per Trial Type and Age Group, **b** shows error rates per Trial Type and Age Group, and **c** shows the proactive behavioral index calculated for reaction times. **d** and **e** Shows the results of the Flanker task. **d** Shows the proportion congruency effect calculated for reaction times and **e** shows the proportion congruency effect calculated for error rates.

## Age differences in shifting

**Plus–Minus task**. The LMM with Box-Cox transformed RTs showed a main effect of Switch ($F(1,297) = 140.58$, $p < 0.001$) and Age Group ($F(3,297) = 40.00$, $p < 0.001$). Participants took longer to complete the switch block ($M = 168$ s) compared to non-switch blocks ($M = 138$ s), indicating a switch cost. Post-hoc comparisons for the main effect of Age Group showed that with age, participants took more time to complete the task. Young adults ($M = 102$ s) were faster than the 60–69 ($M = 136$ s; $t(297) = -3.71$, $p = 0.0014$, $d = -1.32$), 70–79 ($M = 188$ s; $t(297) = -9.41$, $p < 0.001$, $d = -3.34$) and 80+ groups ($M = 186$ s; $t(297) = -8.76$, $p < 0.001$, $d = -3.25$). In addition, the 60–69 group was faster than the 70–79 ($t(297) = -5.81$, $p < 0.001$, $d = -2.02$) and 80+ age groups ($t(297) = -5.31$, $p < 0.001$, $d = -1.93$). No difference was observed between the two oldest age cohorts ($p = 1.00$). We also observed an interaction between Switch and Age Group ($F(3,297) = 8.26$, $p < 0.001$), which was further studied by comparing switch costs between Age

Groups. Switch costs significantly increased with age ($M_{\text{Young adults}} = 12$ s, $M_{60-69} = 28$ s, $M_{70-79} = 46$ s, $M_{80+} = 32$ s), except for the oldest age group. All comparisons were significant ($p < 0.001$, $d \geq 0.55$), except for the difference between the 60–69 and 80+ age group ($p = 0.72$).

The GLMM for the total number of errors showed a significant main effect of Switch ($X^2(1) = 13.57$, $p < 0.001$). Participants made more errors on the switch block ($M = 0.43$ errors or 1.43%) compared to the non-switch blocks ($M = 0.74$ errors or 2.47%), indicating a switch cost. All other effects were not significant ($p \geq 0.062$).

To sum up, older adults needed more time to complete this task than young adults, and switch costs for RT increased with age (except for the oldest age group), whereas no significant differences in errors were observed.

## Age differences in proactive and reactive control

**AX-CPT**. The LMM for RTs showed a significant main effect of Age Group ($F(3,287.56) = 42.19$, $p < 0.001$): with age, participants slowed

**Table 4 | An overview of the results of age differences for all outcome measures per cognitive control component**

| | | | |
|---|---|---|---|
| Inhibition | Go/No-Go | RTs | Young adults are faster than older adults, 60–69 and 70–79 are faster than 80+ |
| | | Go ERR | Young adults make less ERR than older adults<br>60–69 make less ERR than 70–79 and 80+; 70–79 make less ERR than 80+ |
| | | No-Go ERR | Young adults make more ERR than 70–79: **worse response inhibition in young adults** |
| | Flanker | RTs, ERR | Young adults are faster but make more ERR than older adults<br>60–69 and 70–79 are faster than 80+, 60–69 make less ERR than 70–79 and 80+ |
| | | RT CE | Young adults show larger CE than 60–69 and 70–79: **worse interference control in young adults**<br>60–69 show smaller CE than 70–79 and 80+: **most efficient interference control in 60–69** |
| | | ERR CE | Young adults show larger CE than older adults: worse interference control in young adults<br>60–69 and 80+ show larger CE than 70–79: **most efficient interference control in 70–79** |
| Updating | N-Back | RTs | Young adults are faster than older adults, 60–69 are faster than 70–79 and 80+ |
| | | Missed targets | Young adults make less ERR than older adults: **worse updating across age**<br>60–69 make less ERR than 70–79 and 80+: **stagnation in two oldest groups** |
| | | False alarms | Young adults make less ERR than older adults, 60–69 make less ERR than 70–79 and 80+ |
| Shifting | Plus–Minus | RTs, ERR | Young adults are faster than older adults, no difference in ERR<br>60–69 are faster than 70–79 and 80+, no difference in ERR |
| | | RT switch cost | Young adults show smaller switch costs than older adults<br>60–69 show smaller switch costs than 70_79, but 70–79 show larger switch costs than 80+: **worse shifting with age except for oldest age group** |
| | | ERR switch cost | No difference in switch costs between groups |
| Proactive vs. Reactive Control | AX-CPT | RTs | Young adults are faster than older adults, 60–69 are faster than 80+<br>All age groups are slower on AY than BX trials: **all age groups use a proactive control strategy**<br>AY: young adults are faster than older adults, 60–69 and 70–79 are faster than 80+<br>BX: young adults are faster than older adults, no differences in older adults |
| | | ERR | No differences in age groups, for all age groups more ERR on AY than BX trials: all age groups use a proactive control strategy<br>AY: young adults make more ERR than 60–69, BX: no differences |
| | | RT PBI | For all age groups, positive PBI but higher in young vs. older adults: proactive control preference for all age groups, but **stronger proactive control preference in young vs. older adults** |
| | | ERR PBI | No differences between groups, positive PBI: proactive control preference in all groups |
| | Flanker PC | RT, ERR | Young adults are faster but more ERR than older adults, 60–69 and 70–79 faster than 80+ |
| | | RT PCE | Young adults smaller PCE than 60–69, 60–69 show a larger PCE than 70–79 and 80+: **most efficient proactive control in 60–69** |
| | | ERR PCE | Young adults smaller PCE than 70–79 but larger PCE than 60–69<br>60–69 smaller PCE than 70–79 and 80+; 80+ smaller PCE than 70–79: **most efficient proactive control in 70–79** |

Main results are indicated in bold.

*RTs* reaction times, *ERR* errors, *CE* congruency effect, *PBI* proactive behavioral index, *PC* proportion congruency, *PCE* proportion congruency effect.

down on the AX-CPT. Young adults ($M = 448$ ms) were faster than the 60–69 ($M = 597$ ms; $t(287) = -7.28$, $p < 0.001$, $d = -1.01$), 70–79 ($M = 623$ ms; $t(287) = -8.57$, $p < 0.001$, $d = -1.19$) and 80+ groups ($M = 675$ ms; $t(288) = -10.50$, $p < 0.001$, $d = -1.54$). Older adults between 60 and 69 years old were faster than the 80+ age group ($t(288) = -3.71$, $p = 0.0014$, $d = -0.53$). No significant differences were found between the 60–69 and 70–79 groups and between the two oldest Age Groups ($p \geq 0.074$). The main effect of Trial Type ($F(3,274.77) = 415.46$, $p < 0.001$) is further described in the Supplementary results. Regarding the interaction between Trial Type and Age Group ($F(9,274.54) = 3.82$, $p < 0.001$), all pairwise comparisons can be consulted in Supplementary Table 4 and 5. Most importantly, for all Age Groups, AY trials were always significantly slower than BX trials (all $p < 0.001$), indicating the use of a proactive control strategy in all Age Groups. Moreover, for AY trials, young adults were significantly faster than all older adult groups (all $p < 0.001$), and the 60–69 and 70–79 groups were faster than the 80+ group ($p \leq 0.035$). The performance of the 70–79 years olds did not differ significantly from the 60–69 group ($p = 0.061$). For the BX trials, however, young adults were still faster than older adults (all $p < 0.001$), but no significant differences were found between the older adult groups ($p \geq 0.16$).

The GLMM for error rates showed a significant main effect of Trial Type ($X^2(3) = 329.96$, $p < 0.001$), which is further described in the Supplementary

results. We also observed an interaction between Trial Type and Age Group ($X^2(9) = 23.11$, $p = 0.0060$), for which all pairwise comparisons can be consulted in Supplementary Table 6. Importantly, for all Age Groups performance on AY trials was always significantly worse than BX trials (all $p \leq 0.0017$), indicating a proactive control strategy. Moreover, for AY trials, young adults ($M = 10.03\%$) made significantly more errors than the 60–69 group ($M = 5.06\%$; $Z = -3.37$, $p = 0.0042$, $d = 0.74$). There were no other significant differences between Age Groups for the AY trials or other types of trials ($p \geq 0.074$). The main effect of Age Group was not significant ($p = 0.089$).

The one-way ANOVA for the PBI based on RTs showed a significant difference in PBI between at least two Age Groups. Post-hoc paired comparisons showed that young adults ($M = 0.21$) had a higher PBI than older adults aged 60–69 ($M = 0.13$, $p < 0.001$, $d = -0.67$), 70–79 ($M = 0.14$, $p = 0.0014$, $d = 0.59$), and 80+ years ($M = 0.16$, $p = 0.034$, $d = 0.39$), indicating a higher preference for proactive control in young compared to older adults. There were no significant differences in PBI between the older age groups ($p \geq 0.48$). Moreover, there was no difference between groups in PBI based on ERR ($p = 0.74$; $M_{\text{Young adults}} = 0.32$, $M_{60-69} = 0.23$, $M_{70-79} = 0.28$, $M_{80+} = 0.22$). Both for RT and ERR, the PBI for all groups was positive, indicating a preference for proactive control.

**Flanker with proportion congruency manipulation.** Adding the order of the blocks to the models showed that the order of the blocks in this task

can have an impact on the size of the observed PCEs. However, the PCEs remained significant regardless of block order (all $p < 0.001$), and the block order effects did not interact with age. For conciseness, we therefore only included the analyses without block order.

The LMM with Box-Cox transformed RTs showed a significant main effect of Congruency ($F(1,325) = 982.04$, $p < 0.001$): participants reacted slower to incongruent ($M = 659$ ms) than congruent trials ($M = 616$ ms), i.e., a congruency effect. Post-hoc comparisons for the main effect of Age Group ($F(3,293) = 85.11$, $p < 0.001$) showed that with age, participants significantly slowed down on the Flanker task. Young adults ($M = 514$ ms) were faster than the 60–69 ($M = 663$ ms; $t(293) = −11.40$, $p < 0.001$, $d = −1.36$), 70-79 ($M = 671$ ms; $t(293) = −11.89$, $p < 0.001$, $d = −1.42$) and 80+ groups ($M = 735$ ms; $t(293) = −14.78$, $p < 0.001$, $d = −1.86$). The 60–69 ($t(293) = −4.10$, $p < 0.001$, $d = −0.51$) and 70–79 age group ($t(293) = −3.59$, $p = 0.0022$, $d = −0.44$) were faster than the 80+ group. No significant differences were found between the two oldest age groups ($p = 0.95$). The main effect of Block ($F(2,309) = 8.44$, $p < 0.001$), and the significant two-way interactions (Block and Congruency ($F(2,99033) = 196.53$, $p < .001$, i.e., a PCE), Congruency and Age Group ($F(3,325) = 17.42$, $p < .001$)) are further discussed in the Supplementary results. Moreover, we observed an interaction between Block, Congruency, and Age Group ($F(6,99032) = 3.27$, $p = 0.0033$). To study this three-way interaction, the PCE was calculated and compared between Age Groups. The one-way ANOVA for PCE with factor Age Group was significant ($F(3,288) = 5.91$, $p < 0.001$). Post-hoc tests showed that the PCE was larger in the 60–69 age group ($M = 58$ ms) compared to the young adults ($M = 49$ ms, $p = 0.0055$, $d = −0.56$) and 70–79 age group ($M = 48$ ms, $p = 0.0092$, $d = 0.60$) and 80+ group ($M = 50$ ms, $p = 0.028$, $d = 0.43$). All other comparisons were not significant ($p ≥ 0.86$). The interaction between Block and Age Group was not significant ($p = 0.68$).

The GLMM for the errors showed a significant main effect of Age Group ($X^2(3) = 61.97$, $p < 0.001$), showing that young adults made significantly more errors ($M = 4.04\%$) compared to older adults of 60–69 ($M = 1.14\%$, $Z = 9.29$, $p < 0.001$, $d = 1.30$), 70–79 ($M = 1.51\%$, $Z = 7.39$, $p < 0.001$, $d = 1.00$) and 80 years or older ($M = 1.54\%$, $Z = 6.87$, $p < 0.001$, $d = 0.99$). There were no significant differences in errors between the older age groups ($p ≥ 0.17$). All other main effects were not significant ($p ≥ 0.29$). The significant two-way interactions between Block and Congruency ($X^2(2) = 8.26$, $p = 0.016$, i.e., a PCE), Block and Age Group ($X^2(6) = 20.00$, $p = 0.0028$), and Congruency and Age Group ($X^2(3) = 11.92$, $p = 0.0077$) are further discussed in the Supplementary results. The significant three-way interaction between Block, Congruency, and Age Group ($X^2(6) = 17.49$, $p = 0.0076$) was further explored by comparing the PCE between Age Groups. The PCE was larger in the older adults aged 70–79 ($M = 1.79\%$) compared to young adults ($M = 1.24\%$, $p = 0.0012$, $d = −0.51$), older adults aged 60–69 ($M = −0.31\%$, $p < 0.001$, $d = −2.25$), and 80+ ($M = 1.29\%$, $p = 0.0071$, $d = 0.42$). Moreover, the PCE was smaller in the 60–69 group compared to young adults ($p < 0.001$, $d = 2.87$) and the 80+ group ($p < 0.001$, $d = −2.41$). There was no significant difference between the young adults and the 80+ group ($p = 0.98$).

To sum up, no shift from proactive to reactive control was observed. Rather, all groups still preferred a proactive control strategy (indicated by a positive PBI, slower and less accurate AY vs. BX trial performance, and a PCE in each group). However, the preference for proactive control was lower in older compared to young adults in the AX-CPT. The PCE findings on the Flanker task were difficult to interpret, and further research with stronger congruency manipulation is needed to explore proactive control with age.

When controlling for overall RT by calculating the mean RT across all trials for each participant and each task and adding this variable as a fixed main effect in the error rate models, we observed a highly similar pattern of results, with all age group effects still present for all tasks. In addition, comparison of Inverse Efficiency Scores between groups resulted in highly similar patterns as the RT analyses without controlling for accuracy. A detailed description of these speed-accuracy trade-off analyses can be found in the Supplementary Results.

## Discussion

Declines in cognitive control impact the daily life and independence of older individuals and their surroundings. The aim of this study was to provide a comprehensive overview of differences in cognitive control subcomponents with age. With this study, we aimed to bridge current gaps in the field by administering an extensive test battery in large samples of young adults and older adults of different age groups (60–69, 70–79, and 80+), studying inhibition, updating, shifting, proactive, and reactive control.

Results for *inhibition* showed decreases in overall performance for older adults, as indicated by slower RTs and higher omission errors on the Go/No-Go. However, young adults made more commission errors and showed larger congruency effects, which are the two key indices of response inhibition and interference control. This pattern speaks against a general decline in inhibition as found by other cross-sectional studies[11,22–25] and general deficit theories such as the Inhibition Deficit theory[13]. These findings are also not in line with meta-analyses[76,77] that showed no evidence for declines in interference control with aging, but evidence for declines in response inhibition. Instead, the current study confirms findings of Rey-Mermet et al.[78] showcasing better interference control in older compared to young adults. Other studies[35,79] have also shown more commission errors in young compared to older adults. It has been suggested that, rather than worse inhibition in young adults, this could reflect a more cautious response style in older adults, trading speed for accuracy[80,81]. However, even when controlling for speed in the error analyses (see Supplementary results: "Addressing speed-accuracy trade-offs"), commission errors and congruency effects based on errors were still larger in young compared to older adults. Still, Inverse Efficiency Scores were larger in older compared to young adults, showing reduced efficiency on the task with age. The latter seems to indicate a case of general slowing rather than reduced inhibition, as this pattern is similar to what we observed in the RT results. When looking at the results across our older age groups, no differences were found for response inhibition. For interference control, however, we observed larger RT congruency effects for older adults over the age of 70 compared to the 60-69 group. Ferguson et al.[11] also showed declines in interference control with age. These findings argue for differential effects of aging for inhibition subcomponents: response inhibition does not seem to be affected in older age, whereas interference control shows indications of decline with further aging. One potential alternative theoretical explanation for finding declines in interference control but not in response inhibition in the older age groups, albeit speculative, might lie in the fact that cortical responses to different stimuli become less distinctive as people age[82]. This age-related dedifferentiation could be due to the broadening of the tuning curves of category-selective neurons (broadening hypothesis) or to decreased activation of category-selective neurons (attenuation hypothesis). In any case, this phenomenon of neural dedifferentiation would be especially problematic for tasks where irrelevant visual information needs to be suppressed, as was the case in our interference control task. If that irrelevant information is no longer differentiated, but processed to the same extent as the relevant visual information, large congruency effects are to be expected. This could explain the larger decline in interference control compared to response inhibition, where arguably this neural differentiation is less problematic to suppress an upcoming response based on highly salient stimuli in different colors. In line with this, the Inhibition Deficit theory of Hasher and Zacks[13] already proposes that older adults lose the ability to inhibit irrelevant information from distractors and therefore have to deal with more interference. We only observed this aging effect when comparing older age groups, not when comparing young and older adults. However, as inhibition seems to develop up to 21 years old[83] or 25 years old[84], the young adults ($M_{Age} = 18$ years) included in our study may still show impulsive behavior and may not have been the optimal control group.

Regarding *updating*, all analyses clearly showed worse updating with age, resulting in slower RTs, more misses, and more false alarms with age. These findings corroborate earlier evidence for declines in updating with aging[11,22,23,38,85]. Interestingly, this decline seems to stagnate between the ages of 70 and 79, as the two older age cohorts did not show any differences in RTs or error rates.

Similarly, the Plus–Minus task studying *shifting* revealed increasing switch costs with age for RTs, except for the oldest age cohort. No differences with age were detected for error rates. As for updating, these declines for RTs were in line with previous studies[11,22] that indicated worse global switch costs in older vs. young adults. Of note, we were only able to study global switch costs here due to the design of the Plus–Minus task. There is some evidence for declines in global switch costs (i.e., differences between switch and no switch blocks), but age-related changes in local switch costs (i.e., differences between switch and no switch trials within the switch block) remain debated[86]. Future studies should incorporate both global and local switch costs paradigms to obtain a more fine-grained overview of changes with aging in the shifting component.

The results for *proactive and reactive control* for the AX-CPT showed a preference for a proactive control strategy in all age groups, reflected in slower RTs and higher error rates on AY compared to BX trials. This pattern suggests that all participants maintained cue information, therefore invalidly preparing for a target response after seeing an A cue, but being able to anticipate a non-target key press after seeing a B cue[5]. However, the PBI showed a significantly smaller preference for proactive control in older compared to young adults. In addition, younger adults, although being faster, made significantly more AY errors than older adults, indicating that the cue bias was larger in young than older adults or, alternatively, that young adults had worse reactive control to overcome this bias than older adults. The latter corroborates our inhibition findings. These results contrast the hypothesized strong shift from proactive to reactive control[20,28,30]. Although a decline in cue bias and proactive control preference was found, older adults still preferred proactive control similarly to young adults, in line with other research[31–33]. Indeed, as Berger et al.[87] and Xu et al.[88] suggest, older adults may still be capable to use proactive control under the "right" task circumstances. Results for the Flanker task with proportion congruency manipulation showed larger congruency effects in young compared to older adults. This is in line with findings for the inhibition component, showing that reactive control is worse in young compared to older adults. Results for proactive control, indexed by the proportion congruency effect, were unclear. Future research is needed, with a stronger manipulation of proportion congruency (e.g., more extreme differences in proportion congruency conditions), to study age-related differences in proactive control. In addition, others have argued that global switch costs could also reflect proactive control[89–91]. To our knowledge, the Plus-Minus task has not been used yet to assess proactive control. However, as switching is 100% predictable in the switch block of our Plus-Minus task, and thus can be anticipated (cf. also Yu et al.[89]), one could argue that the global switch cost in this task can also be seen as an index of proactive control, which would imply that at least some proactive control aspects declined with older age. Next to the importance of task demands in this field of study, the selection bias in our sample, leading to a rather highly educated sample of older adults, might obscure declines in proactive control present in the more general older adult population. Finally, it could be that behavioral measures are not sensitive enough to capture proactive control declines, and older adults use compensatory strategies to reach good levels of behavioral performance at the expense of over- or under-recruitment of brain activation[92] (for a similar argument in the field of insomnia, see Muscarella et al.[93]).

General age-related slowing was observed across all tasks. This slowing could reflect difficulties with the underlying subcomponent measured by that task. However, this slowing could also be interpreted in light of older adults generally needing more time to process and store information or to respond to a task, in line with the processing-speed theory[14]. Even when controlling for speed-accuracy trade-offs (see Supplementary results: "Addressing speed-accuracy trade-offs"), age-related results patterns remained highly similar, potentially reflecting this general age-related slowing.

To summarize, clear age-related declines were found for shifting and updating, but not for inhibition and proactive control. It becomes clear that there is not one universal decline in cognitive control, as proposed by theories such as the prefrontal-executive hypothesis[9] or the processing-speed

theory[14]. These differential age-related declines depending on the specific subcomponent are in line with other studies[24,26], but we have provided, for the first time to the best of our knowledge, a comprehensive view across multiple subcomponents and in large samples of older adults from different age cohorts, providing an overarching overview of cognitive control and aging. These findings highlight the importance of including multiple cognitive functions and a broad age range when studying cognitive control in older adults.

Regarding limitations and recommendations for future research, this study did not include a middle-age group (28–59), which keeps us from being able to show the full adult lifespan changes. For this reason, we chose not to treat age as a continuous variable, but rather to use separate age groups. Future studies including this age group would be useful[11]. Moreover, the young adults included in this study were mainly undergraduates with a mean age of 18 and predominantly female. A more balanced sample should be included in future studies. In addition, as in most aging research, participants motivated to enroll in the study could already have been cognitively stronger, inducing a selection bias[3]. We tried to keep the inclusion criteria as inclusive as possible, and conducted the participant recruitment very extensively and intensively. However, older adults were required to be cognitively healthy (cf. MoCA and GDS inclusion criteria and self-reported diseases), were mainly highly educated, and were living independently at home, which could have led to an underestimation of cognitive control declines with age. Indeed, a health survey from Sciensano in Belgium in 2023[94] showed that 40% of the older adults aged 65 years or older reported having a chronic disease. For example, around 15% of older adults reported having diabetes, and about 8.6% reported having severe heart disease (averaged over sex). This is in clear contrast to the reported 0.87% for both diseases in our study. In addition, Van der Heyden et al. (2025) reported that almost 6% of adults aged 65 years or older reported having suffered from depression during the past year. Again, this is not in line with our sample, where a score of six or lower on the GDS-15 was used as an inclusion criterion. Moreover, the demographic information in our study was based on self-report, and the relevant question pertaining to diseases only mentioned diseases that could have an influence on cognitive functioning. Hence, not all diseases may have been reported by the participants. Moreover, motivation may have differed between age groups. Whereas young adults participated in the study for course credits, older adults voluntarily participated in the study. This could have further increased a bias. Future research should include motivation measures to account for this. In addition, as already discussed above, future studies should administer a shifting task that also allows to study local switch costs, which was not the case in the current Plus–Minus task. Moreover, as only one speed and accuracy index is obtained per list in the classic Plus-Minus task, error variance might have been large here. Providing multiple trials could improve the interpretability of age effects for shifting. Finally, for tasks with strict response deadlines, such as the Go/No-Go and N-Back, the same response deadline was used across all age groups. However, an adaptive response deadline could be an interesting approach in future research to make the experienced time pressure more comparable across participants and age groups.

To conclude, this study is one of the first, to the best of our knowledge, providing a comprehensive view of cognitive control subcomponents in aging. By administering an extensive test battery in young adults and different cohorts of older adults with large sample sizes, this study shows that heterogeneity not only exists in age-related changes in cognitive control between older adults[12,95,96], but also between components and tasks, and even within a cognitive control subcomponent (cf. inhibition). Our results demonstrate that with aging, not all cognitive control functions decline, and that some are preserved or even improved compared to young adults, offering a more nuanced view on cognitive decline in aging. It is thus crucial to include multiple components in future aging studies, as there is no domain-general decline in cognitive control. To expand these findings, longitudinal studies are necessary to track trajectories of changes in cognitive functions over time and age, accounting for intra- and inter-individual differences. Moreover, protective and risk factors for cognitive decline, such

as education, depressive symptoms, and social network, should be included, as the concept of cognitive reserve[97] already showed the beneficial effect of education and an active lifestyle on cognitive functioning in later life. Theories such as the compensation-related utilization of neural circuits hypothesis (CRUNCH[98]) or the revised Scaffolding Theory of Aging and Cognition (STAC-R[99]) may be more suitable for explaining age-related differences than general deficit theories, considering multiple levels of processing at which older adults can compensate for age-related declines in cognitive functioning, although these frameworks are not specific to cognitive control. Neuroimaging or EEG studies could be interesting in that regard, possibly exposing more subtle neural differences or strategies when exerting cognitive control that differ with older age, going beyond behavioral performance. Our data also allows exploration of other outcome measures for cognitive performance, such as intra-individual variability[100,101]. In the Supplementary results, we already showed an increase in intra-individual variability in task performance with age, indexed by larger standard deviations of reaction times in older adults. We invite other researchers to further explore the data on the Open Science Framework[49]. Linking this behavioral intra-individual variability to these neural measures and to possible protective and risk factors could further shed light on functional stability and noise with aging[100,101]. After all, as this study shows, not all is bad with older age.

## Data availability
The anonymized data that support the findings of this study and the tasks used in this study are available in the Open Science Framework[49] (OSF, https://doi.org/10.17605/OSF.IO/VBH58). All figures are based on these source data, available for each task in the repository.

## Code availability
The R code used to generate the results and the figures is available in the Open Science Framework[49] (OSF, https://doi.org/10.17605/OSF.IO/VBH58). All analyses were performed in R[73] (v4.2.1) and RStudio[74] (v2024.4.2.764), with the lme4 package[72] (v1.1.31) and the emmeans package[75] for post-hoc tests (v1.8.5).

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

## Acknowledgements

The authors want to thank all participants in this study, especially all older adults participating and welcoming us into their homes. Thank you as well to all students and colleagues involved in the recruitment and data collection: Isabel Meskens, Charlotte Vankerkhoven, Lisa Kellens, Hanne Claes, Elisabeth Stockman, Amber Hofmans, Fleur Janssens, Yana Baert, Rino Boisschot, Noémie Peeters, Ellaline Cami, and Febe Demeyer. This study was funded by the Research Foundation Flanders (FWO; grant number 11J1221N) and by KU Leuven Internal Funds (grant number C14/21/046). Author Céline R. Gillebert is supported by a Methulasem grant (METH/24/003).

## Author contributions

S.D.P., C.R.G., E.D. and E.V.D.B. designed the study; S.D.P. collected the data together with the help of students (cf. acknowledgements); S.D.P., H.S., and E.V.D.B. analyzed the data; S.D.P. and E.V.D.B. wrote the manuscript; C.R.G., E.D., H.S. and E.V.D.B. provided feedback on previous drafts of the manuscript.

## Competing interests

The authors declare no competing interests.
