## [Transparent Peer Review file · Communications Medicine]

A Comprehensive Study of Cognitive Control in Healthy Aging

Corresponding Author: Dr Sarah De Pue

Version 0:

Reviewer comments:

Reviewer #1

(Remarks to the Author)

The current manuscript investigated the age-related differences in multiple cognitive controls in large samples of younger and older adults. Comprehensive assessments of different control mechanisms were conducted. Older adults from different age groups were included. The literature review was well conducted. Overall, this is a high-quality study. I only have a few comments.

1. The authors treated Miyake's and Braver's definitions of cognitive control as mutually exclusive and independent. This is not appropriate, as similar tasks (esp. task-switching) have been used to test and validate both theories. In task-switching literature, global switch cost measures proactive control, while local switch (or mixing) cost measures reactive control. The AX-CPT task measures the "preference" for using one control over the other, but not necessarily how well one performs using either control. The current AX-CPT task has only 20 trials of AY and BX trials across two blocks—a number too small to assess "performance." From the plus-minus task, older adults definitely performed worse in global switch cost (proactive control). There was no local switch cost or reactive control task. The authors should reframe their arguments conclusion that there was "no significant decline in proactive control". There was no difference in "preference," but "performance" declined.
2. The authors stated that accuracy cut off was set at 25%. Did this mean participants with less than 25% accuracy on a task were dropped from the analysis? Also 25% was too lenient for these tasks, I suggest changing the criterion to 50% and re-run analyses.
3. For the LMMs, can the authors present an example equation somewhere in the methods or supplementary, and the covariates (if any)?

Reviewer #2

(Remarks to the Author)

The study by De Pue et al aimed to assess if decline in executive functioning and cognitive control is general or specific in aging, and is expressed in a similar way in three cohorts of older adults from 60 to 95 years of age. A comprehensive battery of tasks assessing updating, shifting and inhibition, as well as measures of proactive/reactive control was administered to 75 young and 231 older adults. Results showed better inhibitory control and interference resolution in the youngest group of older adults than in young people, while updating and shifting abilities decrease with age. These findings show that executive decline in healthy is not general and may have different trajectories according to sub-processes.

This is a well conducted study whose strengths are the large number of tasks administered, the use of measures and indexes theoretically grounded and a large sample of older participants, with several age groups. Statistical analyses are adequate regarding the objective and level of details provided allow reproducibility. The discussion of results is fair, allowing to relate discrepancies in previous results reported in the literature. The manuscript is well written and includes the recent and relevant literature. Data reported will help in the understanding of age-related changes in cognitive control, in the context of current theoretical frameworks..

The following comments may help to go further in the interpretation of the results

A main finding of the study is the presence of a dissociation in aging between no decline in interference control and decline in response inhibition. The proposed explanation is different trade-off speed / accuracy across groups. Similarly, it seems that older adults remain able to use proactive control, contrary to what was observed in previous studies. Here, the "right" task circumstances is proposed to explain discrepancies in the literature. While I agree with the explanations proposed by

the authors. I would nevertheless suggest emphasizing theoretical interpretation, if they exist.

Another explanation that may deserve a short discussion is the motivation of participants to perform the task. Young subjects were credited for participating in the study that is different from older subjects volunteering to participate. The authors mention the use of NASA-TLX for measuring subjective task-related effort. Even if this measure does not correspond to motivation, it may be a prospective way to discuss how effortful the tasks that are impacted (or not) by aging. Obviously, we cannot infer motivation for subjective effort put in a task but capitalizing on these data may add to the discussion and future studies.

The young group has higher scores on the GDS-15. (see Table 2). Are similar results than those reported in the manuscript obtained when score at GDS is controlled for in the analyses? If the effects are still present, this is in favor of the robustness of the interesting results reported. More generally, is there an association between depression score and subjective effort reported. If there is an association, is it group-specific or group independent?

Finally, to measure proactive and reactive control in the context of the flanker task, the proportion of incongruent trials was manipulated across three blocks, whose order of presentation was counterbalanced across participants. However, it cannot be excluded that facing first the block inducing proactive (vs reactive) control influences the kind of control implemented in subsequent blocks. Did the authors check if implementation of proactive / reactive control vary according to the proportion of incongruent trial in the first block ?

Reviewer #3

(Remarks to the Author)

The authors present a manuscript in which they examine changes of subcomponents of cognitive control at different ages. This is an interesting and timely study with a relatively large number of participants. The task selection to analyze cognitive control, the methods, data analysis and result presentation sound well. However, there are some missing information, methodological problems and problems with the data analysis that may lead to incorrect conclusions. Overall, there is some space for improvement. I will go through the manuscript and comment on the problems in the order in which they occur. Overall, the authors used 4 groups of participants with young 18-27 years old and older beginning with the age of 60. That is, there is a large age gap in the middle-age between 27 and 60 years preventing conclusions about cognitive changes across the adult lifespan. It is well known that cognitive changes begin in the early fifties due to progressive thinning of the cortex, mainly at prefrontal areas responsible for executive and cognitive control functions.

The difference in education between age groups is surprising. Young adults had a lower number of years than the older ones or even the oldest group. In most European countries, the education level after the second world war (80 or 70 years ago) was only rudimentary and mostly elementary school was finished. Please explain this discrepancy in more detail and include the range of education years in the table. Secondly, the ratio of females was larger in the young than the older groups that as well as higher MoCA scores in the young-old than old-old group also needs some explanation. Please compare statistically the groups regarding GDS-15 and years of education.

The authors obtained detailed sociodemographic questions which is positive. Is there any information about marital status, loneliness, quality of life, regular physical or cognitive activity, nutrition etc., i.e., factors that are important for cognitive functioning in older age? From the supplementary table 2 it was surprising to see that there were only a few diseases in the older groups. Depression symptoms, diabetes, heart or cardiovascular disease are common in older population but not in the present sample. Do the authors have an explanation for this or were they a priori excluded from the sample? The most important question is how representative is the sample compared to a general population. It would be helpful if the authors could provide some reference parameters to assess the representativity.

Methods: cognitive test battery. The tasks and questionnaires were applied at home. Where there an experimenter who accompanied the session, or did the participants conduct the test battery self-paced? If so, how did the authors control compliance with the test program and control distraction and other individual parameters? Are different laptops / PC used for data collection? This is important for the comparability of data sets. Please discuss in the section Limitations of the study.

It seems that in the Go/NoGo task there was no time pressure to avoid waiting strategy. The core of the task is to press a response button as fast as possible to provoke failures of inhibitory control (i.e. to produce no-go errors). The problem with this task is that the same response deadline of 500 ms was applied in all age groups and it is known that older participants show longer RTs than younger ones (age-related slowing). A much better solution would be an adaptive deadline individually computed for each participant based on mean RTs (or mean + 1SD). This would ensure comparable time pressure in young and in old participants. The supposed shift of strategy between young and old could be due to different feelings of time pressure. This methodical shortcoming having generally huge effects on the cognitive performance and conclusion about age-related differences in inhibitory control should be discussed in the section Limitations of the study.

The authors used RTs and error rates as dependent variables. However, in cognitive tasks comparing different age groups speed-accuracy tradeoff may occur. This is apparent for example in the Go/NoGo tasks, where the young group show the highest error rates and fastest responses. On the other hand, there is age-related slowing, as well lower error rates in older age, mainly in difficult e.g., incongruent or switch conditions. How can the authors exclude the speed-accuracy tradeoff?

There are some ways to solve the problem and to weight both parameters equally. For example, by computing inverse efficiency scores (IES; Townsend and Ashby, 1983). Alternatively, there is the possibility to compute the drift rate obtained using a drift diffusion model (Ratcliff, 1978). Both parameters, IES and drift rate, reflect a compound of speed and accuracy and would make it possible to compare overall performance between age groups.

How long was the response deadline in the Flanker and the AX-CPT tasks? Overall, in both tasks applied in different age groups there is the same problem with speed deadlines, as mentioned above.

It is unclear what does mean the 30% hit rate in the N-back task. Is it the rate of targets? Please provide the number for targets and non-targets for each task. Please rename the error types in the common ones (omissions or missed targets) and false alarms (response to non-target).

In the task switching, were the single task blocks averaged and then subtracted from the switch block? Were the trial by trial RTs recorded and analyzed or the total time to perform a block? This is unclear. In general, there is a lack of theoretical

background for the task switching. Moreover, it is unclear why both global (mixing) and local switch costs were not reported. This specific design only became clear during the discussion and needs to be introduced earlier.

Were the results of the digit span correlated with the cognitive tasks?

Inhibition was studied using 47C block of the Flanker task. Why? Thus, the overall congruency effect was rather small (20 ms). Inhibitory control is mostly required in blocks with rare targets (87C) and the congruency effect should be much larger. The small congruency effect (together with large data variability in older age, particularly in difficult conditions) could be the reason for no significant difference between the young and the 80+ group. This may lead to the wrong conclusion that older groups show superior cognitive control compared to the younger group. Additionally, the authors interpreted the 47C data and suggested that more extreme differences in proportion congruency conditions are needed. This procedure is unclear. In Fig 1. The Flanker congruency effect was measured in milliseconds and in %. But there are some negative values. Specifically, almost all values in the age group 70-79 are negative. Why?

Are the results in the section "Flanker with proportion congruency manipulation" across all 3 blocks? As the interaction Age x Block x Congruency was significant, I would suggest reporting the results of the Age x Congruency interaction for each block separately. In other words, the CE should be compared between the age groups for each block separately. This allows to see how the CE changes across the age depending on the ratio of incongruent trials and would increase the transparency. When looking at the mean RTs in Figures 1 and 2 there is an apparent increase of RT variability with age. My suggestion is to use an appropriate measure and to analyze the variability as an important parameter of cognitive performance in age that allow conclusions about functional stability or "noise" in the brain. (c.f. Hultsch & MacDonald, 2004; MacDonald et al., 2006). This seems to be important for the present study as acknowledged in the Conclusions.

Additionally, due to the age-related slowing and high RT-variability the difference values such as CE are less valid or show small or even negative values (longer RTs in congruent than incongruent trials).

It is unclear why X² test was used instead a F-test. Why did the authors use binomial error distribution instead of individual mean error rates? Is box-cox transformation similar to Log10 transformation to transform continuous data to normal distribution? Did the authors prove that the data is indeed normally distributed after the transformation?

Ref:

Hultsch, D.F., MacDonald, S.W.S., 2004. Intraindividual variability in performance as a theoretical window onto cognitive aging. In: Dixon, R.A., et al. (Eds.), *New Frontiers in Cognitive Aging*. Oxford University Press, Oxford, pp. 65–88.

MacDonald, S.W.S., Nyberg, L., Bäckman, L., 2006. Intra-individual variability in behavior: links to brain structure, neurotransmission, and neuronal activity. *Trends Neurosci.* 29, 474–480.

Ratcliff, R., 1978. A theory of memory retrieval. *Psychol. Rev.* 85, 59–108.

Townsend, J.T., Ashby, F.G., 1983. *Stochastic Modeling of Elementary Psychological Processes*. Cambridge University Press, Cambridge.

Version 1:

Reviewer comments:

Reviewer #1

(Remarks to the Author)

The Authors have addressed my concerns. I have no further comment.

Reviewer #2

(Remarks to the Author)

The authors have responded satisfactorily to my comments and requests for clarification (apologies for my sentence with typo about your results in my previous report).

Reviewer #3

(Remarks to the Author)

The reviewer appreciates the effort the authors invested in the revision of the manuscript. The authors responded adequately to almost all my comments and discussed the problems with the demographic data and representativeness of the sample, but the section should be named "Limitations" instead of "recommendation for future research".

However, in my opinion the issue with the control of the speed-accuracy tradeoff is not completely solved and the reviewer is still not fully convinced. This is an important problem as the design of the study and the data analysis strategy may considerably impact the conclusions. Regarding Go/NoGo task the strategy to perform the task in young and older adults seems to be different. Nevertheless, the conclusion of the findings was that there is no inhibitory decline on older age. This may lead to wrong conclusions such as "This pattern speaks against a general decline in inhibition as found by other cross-sectional studies 11,22–25 and general deficit theories such as the Inhibition Deficit theory" which is simply not true. This is a strong conclusion and is rather speculative as the data were not analyzed in the most common way using, for example mixed ANOVAs with aggregated data. The additional results can be included in the supplementary material.

I think this was a misunderstanding about computing the IE Score. IES reflects the relationship between the mean individual speed in correct Go trials and the ratio of commission errors. The simplest way to test speed-accuracy trade-off is to correlate RTs and Error rates across all participants or separately for each age group. Nevertheless, I encourage the authors to compute the individual inverse efficiency score integrating both behavioral parameters and to add this to the manuscript as a

simple but informative index of performance:

The computation for the values Go/NoGo task of the averaged values is: Mean RT in Go trials / (100 – mean ratio of false alarms in NoGo trials) * 100

For the 2-back task: Mean RT in correct target/ (100 – mean ratio of missed targets) * 100

For the AX-CPT (either across all conditions or separately for each condition): Mean RT in correct trials / (100 – mean ratio of errors) * 100

Finally, in the plus minus task, the authors may provide individual RT in the simple blocks and the switch block / (100 – mean ratio of errors in the corresponding block) * 100

Regarding switching task: The reviewer has understood that the task only provides one speed value per block. However, the determination of a total performance time contains large error variance. This can be significantly reduced by using multiple trials in each condition. The more trials, the lower the variability and error variance in the data. If the task was presented on a trial-by-trial basis, why were the individual RTs not analyzed since each trial was completed with a button press? Or did the participants give the answers verbally? This is unclear. Please discuss this point in the section “Limitations”

Negative values in the flanker task (incongruent – congruent) indicate slower responses in congruent vs. incongruent trials in the oldest group. This supports the age-related slowing and increasing intraindividual variability of speed in older age that may blur the differences between congruent and incongruent trials. In my view this is a clear indication for different strategies to perform the task between young and old adults and should be extensively discussed. Additionally, my suggestion to analyze and to illustrate the individual variability is still valid and should be added to the manuscript or at least supplementary material.

Version 2:

Reviewer comments:

Reviewer #3

(Remarks to the Author)

The authors addressed all my concerns. I do not have any further comments and I endorse this manuscript for publication.

Reviewer #1:

The current manuscript investigated the age-related differences in multiple cognitive controls in large samples of younger and older adults. Comprehensive assessments of different control mechanisms were conducted. Older adults from different age groups were included. The literature review was well conducted. Overall, this is a high-quality study. I only have a few comments.

1. The authors treated Miyake's and Braver's definitions of cognitive control as mutually exclusive and independent. This is not appropriate, as similar tasks (esp. task-switching) have been used to test and validate both theories. In task-switching literature, global switch cost measures proactive control, while local switch (or mixing) cost measures reactive control. The AX-CPT task measures the "preference" for using one control over the other, but not necessarily how well one performs using either control. The current AX-CPT task has only 20 trials of AY and BX trials across two blocks—a number too small to assess "performance." From the plus-minus task, older adults definitely performed worse in global switch cost (proactive control). There was no local switch cost or reactive control task. The authors should reframe their arguments conclusion that there was "no significant decline in proactive control". There was no difference in "preference," but "performance" declined.

We fully agree that these frameworks are not strictly exclusive or independent, but we argue that both frameworks can inform us on cognitive control in healthy aging, and together can provide a more comprehensive view. Whereas the Unity and Diversity framework focuses on core cognitive control subcomponents, the Dual Mechanism of Control framework distinguishes cognitive control strategies based on the time scale they operate on. Thus, both frameworks offer a different perspective. We have added a clarification of this in the Introduction.

Introduction, p. 4: *"While these two frameworks are not mutually exclusive, they offer different perspectives on cognitive control in healthy aging."*

To be able to incorporate these perspectives, we selected tasks for each subcomponent that are often used in the literature. Most studies investigating proactive and reactive control, especially in aging, use either the AX-CPT (e.g., Braver & Barch, 2002; Braver, 2012; Paxton et al., 2006; Kray et al., 2015) or conflict tasks with a proportion congruency manipulation (e.g., Xiang et al., 2016; Hunter et al., 2023), which explains our task selection for these cognitive control components. We refer the reviewer to the manuscript for the references.

The AX-CPT, and especially the PBI we discuss in the results, indeed offers an indication of a preference for either proactive or reactive control. However, performance on AY and BX trials can still inform us about proactive/reactive control performance in young vs. older adults. We have clarified this in the Methods as well.

Methods, p. 25: *"The PBI can inform us about a preference for either proactive or reactive control, whereas performance on AY and BX trials can inform us about proactive/reactive control performance in young vs. older adults."*

Although older adults generally slowed down with age, they did not make more errors on AY trials than young adults, indicating no declines in proactive control. Similarly, no clear evidence for declines in the PCE, a proactive control index in the Flanker task with proportion congruency manipulation, was found in older compared to young adults. Therefore, we

remain cautious in claiming a decline in proactive control performance in older age. It could indeed be that global switch costs offer a window into proactive control, which in the current study would indicate that proactive control may decline in older adults. We have added this to the discussion as a cautionary note, suggesting that maybe proactive control is indeed not as intact as we previously suggested.

Discussion, p. 17: *“Others have argued that global switch costs could also reflect proactive control (Yu et al., 2017; Capizzi et al., 2020; Karayanidis et al., 2011). To our knowledge, the Plus-Minus task has not been used yet to assess proactive control. However, as switching is 100% predictable in the switch block of our Plus-Minus task, and thus can be anticipated (see also Yu et al., 2017), one could argue that the global switch cost in this task can also be seen as an index of proactive control, which would imply that at least some proactive control aspects declined with older age.”*

2. *The authors stated that accuracy cut off was set at 25%. Did this mean participants with less than 25% accuracy on a task were dropped from the analysis? Also 25% was too lenient for these tasks, I suggest changing the criterion to 50% and re-run analyses.*

Participants were excluded from the analysis of a specific cognitive control subcomponent if they had fewer than 25% valid data points relative to the total number of trials in each condition of that task. We have reran the analyses with a 50% cut-off, and results remained highly similar. However, this more strict criterion excludes a large number of older adults, especially from the oldest age groups for the N-Back, Plus-Minus and AX-CPT. As this significantly reduces statistical power, and increases the selection bias that already occurred (cf. Discussion, p. 18), we opted to retain the 25% criterion. We did add a footnote to the Method section.

Methods, p. 28: *“When adopting a more strict cut-off of 50%, results remained highly similar. As this significantly reduced sample sizes, especially for the oldest age group, the 25% criterion was retained.”*

3. *For the LMMs, can the authors present an example equation somewhere in the methods or supplementary, and the covariates (if any)?*

Below the reviewer can find an example equation of the LMM for the reaction times on the 47C block of the Flanker, used to study inhibition. This equation can be extended to other models in the manuscript as well. No covariates were added in the analyses.

Level 1: $Y_{ij} = \beta_{0j} + \beta_1 \text{Contrast1}_{ij} + \beta_2 \text{Contrast2}_{ij} + \beta_3 \text{Contrast3}_{ij} + \beta_{4j} \text{Congruency}_{ij} + \beta_5 \text{Contrast1} * \text{Congruency}_{ij} + \beta_6 \text{Contrast2} * \text{Congruency}_{ij} + \beta_7 \text{Contrast3} * \text{Congruency}_{ij} + \varepsilon_{ij}$

Level 2: $\beta_{0j} = \beta_0 + u_{0j}$

$$\beta_{4j} = \beta_4 + u_{4j}$$

In this linear mixed effects model, Y_{ij} represents the reaction time for participant j on the i -th trial, modeling trial-level variations in reaction times. The level 1 model includes fixed effects for the three age group contrasts (i.e., Contrast 1: young adults vs. the 80+ group; Contrast 2: the 60-69 group vs. the 80+ group; and Contrast 3: the 70-79 group vs. the 80+ group), congruency (i.e., congruent vs. incongruent), and their interaction terms. The term ε_{ij} represents the residual error for trial i of participant j , capturing trial-level variability that could

not be explained by the fixed and random effects. The intercept and the slope for congruency are allowed to vary across participants, as specified in level 2. Specifically, each participant has a unique intercept β_{0j} and a unique slope for congruency β_{4j} , modeled as deviations (i.e., u_{0j} and u_{4j} , respectively) from the overall intercept β_0 and slope for congruency β_4 .

We have added this to the supplementary materials (under Supplementary Methods) and have referred to this in the manuscript.

Methods, p. 29: *“An example equation of the LMM for the reaction times for one of the tasks is provided in the Supplementary Methods, which can be extended to other models in the manuscript as well.”*

Reviewer #2:

The study by De Pue et al aimed to assess if decline in executive functioning and cognitive control is general or specific in aging, and is expressed in a similar way in three cohorts of older adults from 60 to 95 years of age. A comprehensive battery of tasks assessing updating, shifting and inhibition, as well as measures of proactive/reactive control was administered to 75 young and 231 older adults. Results showed better inhibitory control and interference resolution in the youngest group of older adults than in young people, while updating and shifting abilities decrease with age. These findings show that executive decline in healthy is not general and may have different trajectories according to sub-processes.

This is a well conducted study whose strengths are the large number of tasks administered, the use of measures and indexes theoretically grounded and a large sample of older participants, with several age groups. Statistical analyses are adequate regarding the objective and level of details provided allow reproducibility. The discussion of results is fair, allowing to relate discrepancies in previous results reported in the literature. The manuscript is well written and includes the recent and relevant literature. Data reported will help in the understanding of age-related changes in cognitive control, in the context of current theoretical frameworks..

The following comments may help to go further in the interpretation of the results

1. A main finding of the study is the presence of a dissociation in aging between no decline in interference control and decline in response inhibition. The proposed explanation is different trade-off speed / accuracy across groups. Similarly, it seems that older adults remain able to use proactive control, contrary to what was observed in previous studies. Here, the “right” task circumstances is proposed to explain discrepancies in the literature. While I agree with the explanations proposed by the authors. I would nevertheless suggest emphasizing theoretical interpretation, if they exist.

With regards to interference control and response inhibition, we actually found the opposite in our manuscript than what the reviewer indicates: no differences in response inhibition between the older age groups, whereas interference control declined across the oldest age groups. Moreover, inhibition performance was better in older compared to young adults. These observations of no decline in response inhibition, nor in proactive control are not in line with the theoretical frameworks we departed from, as indicated in the discussion (p. 15). However, the reviewer is correct that we can of course also interpret these findings from other perspectives. Crucially, these two components of inhibition are clearly distinguishable (Rey-Mermet et al., 2018; Friedman & Miyake 2004), also at the neural level (Long et al., 2022). Other studies (in adolescents for example) have also shown that these processes

only show low to moderate correlations across developmental age spans (Khng & Lee, 2014). Hence, opposite findings as the ones we observed for these components are not impossible. One potential alternative theoretical explanation for our pattern of results, albeit speculative, might lie in the fact that cortical responses to different stimuli become less distinctive as people age (Park et al., 2012). This age-related dedifferentiation could be due to the broadening of the tuning curves of category-selective neurons (broadening hypothesis) or to decreased activation of category-selective neurons (attenuation hypothesis). In any case, this phenomenon of neural dedifferentiation would be especially problematic for tasks where irrelevant visual information needs to be suppressed, as was the case in our interference control task. If that irrelevant information is no longer differentiated, but processed to the same extent as the relevant visual information, large congruency effects are to be expected. This could explain the larger decline in interference control compared to response inhibition, where arguably this neural differentiation is less problematic to suppress an upcoming response based on highly salient stimuli in different colors. In line with this, the Inhibition Deficit theory of Hasher & Zacks (1988) already proposes that older adults lose the ability to inhibit irrelevant information from distractors and therefore have to deal with more interference. We only observed this aging effect when comparing older age groups, not when comparing young and older adults. However, as inhibition seems to develop up to 21 years old (Forte et al., 2024) or 25 years old (Knežević & Marinković, 2017), the young adults ($M_{Age}=18$ years) included in our study may still show impulsive behavior and may not have been the optimal control group.

We have added this potential alternative theoretical explanation to the Discussion.

Discussion, p. 15-16: "One potential alternative theoretical explanation for finding declines in interference control but not in response inhibition in the older age groups, albeit speculative, might lie in the fact that cortical responses to different stimuli become less distinctive as people age (Park et al., 2012). This age-related dedifferentiation could be due to the broadening of the tuning curves of category-selective neurons (broadening hypothesis) or to decreased activation of category-selective neurons (attenuation hypothesis). In any case, this phenomenon of neural dedifferentiation would be especially problematic for tasks where irrelevant visual information needs to be suppressed, as was the case in our interference control task. If that irrelevant information is no longer differentiated, but processed to the same extent as the relevant visual information, large congruency effects are to be expected. This could explain the larger decline in interference control compared to response inhibition, where arguably this neural differentiation is less problematic to suppress an upcoming response based on highly salient stimuli in different colors. In line with this, the Inhibition Deficit theory of Hasher & Zacks (1988) already proposes that older adults lose the ability to inhibit irrelevant information from distractors and therefore have to deal with more interference. We only observed this aging effect when comparing older age groups, not when comparing young and older adults. However, as inhibition seems to develop up to 21 years old (Forte et al., 2024) or 25 years old (Knežević & Marinković, 2017), the young adults ($M_{Age}=18$ years) included in our study may still show impulsive behavior and may not have been the optimal control group."

With regards to proactive control, task context and sample characteristics seem essential. An increasing number of studies, including ours, is not observing declines in proactive control and a preference for proactive control extending to older age (Xiang et al., 2016; Manard et al., 2017; Kray et al., 2015; Hu et al., 2019). As such, most studies refer to the importance of task demands in this field of study (Berger et al, Xu et al). Furthermore, the selection bias in our sample, leading to a rather highly educated sample of older adults, might obscure declines in proactive control present in the more general older adult population. Finally, it

could be that behavioral measures are not sensitive enough to capture proactive control declines, and older adults use compensatory strategies to reach good levels of behavioral performance at the expense of over- or under-recruitment of brain activation (Cabeza et al., 2012; for a similar argument in the field of insomnia, see Muscarella et al., 2019). We have stressed this more in the discussion.

Discussion, p. 17-18: *“These results contrast the hypothesized strong shift from proactive to reactive control. Although a decline in cue bias and proactive control preference was found, older adults still preferred proactive control similarly to young adults, in line with other research. Indeed, as Berger et al. and Xu et al. suggest, older adults may still be capable to use proactive control under the “right” task circumstances.”; “Next to the importance of task demands in this field of study, the selection bias in our sample, leading to a rather highly educated sample of older adults, might obscure declines in proactive control present in the more general older adult population. Finally, it could be that behavioral measures are not sensitive enough to capture proactive control declines, and older adults use compensatory strategies to reach good levels of behavioral performance at the expense of over- or under-recruitment of brain activation (Cabeza et al., 2012; for a similar argument in the field of insomnia, see Muscarella et al., 2019).”*

2. Another explanation that may deserve a short discussion is the motivation of participants to perform the task. Young subjects were credited for participating in the study that is different from older subjects volunteering to participate. The authors mention the use of NASA-TLX for measuring subjective task-related effort. Even if this measure does not correspond to motivation, it may be a prospective way to discuss how effortful the tasks that are impacted (or not) by aging. Obviously, we cannot infer motivation for subjective effort put in a task but capitalizing on these data may add to the discussion and future studies.

We agree that motivation possibly differed between age groups, especially between young and older adults in general. As the reviewer suggests, indeed, young adults could have been less motivated as they participated for course credits, whereas older adults voluntarily took part in the study.

We have computed a general mean score of experienced effort, across all tasks. Higher scores reflect higher experienced effort as indicated by the participant. Effort differed between at least two age groups ($F(3,302)=3.06, p=.029$). Descriptive statistics showed a trend of higher subjective effort reported in young adults ($M=52.53$) and 80+ group ($M=51.78$) compared to the 60-69 ($M=46.34$) and 70-79 group ($M=44.89$). However, post-hoc comparisons with Tukey multiple comparison correction did not show any significant differences in subjective effort between the groups ($p>.064$). However, the NASA effort scale indeed is likely not the best index of motivation, but rather tracks task difficulty (see Violato et al., 2025). In that respect, it would not be surprising that decreased task performance in the oldest adults is accompanied by higher experienced effort. Future studies should incorporate more reliable motivation measures, and we have stressed the potential role of motivational differences now in the Discussion.

Discussion, p. 19: *“Moreover, motivation may have differed between age groups. Whereas young adults participated in the study for course credits, older adults voluntarily participated in the study. This could have further increased a bias. Future research should include measures of motivation to account for this.”*

3. The young group has higher scores on the GDS-15. (see Table 2). Are similar results than those reported in the manuscript obtained when score at GDS is controlled for in the analyses? If the effects are still present, this is in favor of the robustness of the interesting results reported. More generally, is there an association between depression score and subjective effort reported. If there is an association, is it group-specific or group independent?

First of all, we want to stress that for this study, we have already excluded participants with scores above 6 on the GDS-15, thereby only including participants who report low depressive symptomatology (see Methods p. 21). Although young adults score significantly higher, this is still below the clinical cut-off.

Second, when looking at correlations between depressive symptomatology and task performance in the total sample, a small positive correlation was only observed for the Go/No-Go task ($r=0.15$, $p=.013$). This indicates that, overall, depressive symptomatology was not strongly related to task performance in our study (which might not be surprising, based on the use of GDS-15 as exclusion criterion).

Third, when excluding participants who score higher than 4 on this questionnaire ($N=27$), results remained highly similar. After this further exclusion, no significant differences in scores on the GDS-15 between groups were detected anymore ($p=.062$). However, as this exclusion significantly decreased the sample sizes, especially in the youngest adult group ($N_{\text{young adults}}=57$, $N_{60-69}=76$, $N_{70-79}=80$, $N_{80+}=66$ compared to originally $N_{\text{young adults}}=75$, $N_{60-69}=80$, $N_{70-79}=82$, $N_{80+}=69$ before excluding based on GDS-15), and since we already used depressive symptoms as an exclusion criterion, we opted to include these participants in the analyses to retain sufficient power. As analyses are already very complex and some models already did not converge in the original analyses, especially for the Flanker task, and since results remain highly similar when excluding those with a higher score on the GDS-15, we opted not to include depressive symptoms as a covariate. We have added a footnote to the Results.

Results, p. 7: *“When excluding participants who scored higher than 4 on the GDS-15, no significant differences in depressive symptoms were detected anymore between groups ($p=.065$). However, as the pattern of results remained highly similar and sample size was significant decreased because of this additional exclusion, we opted to include all participants in the study.”*

Finally, regarding correlations between depression scores and subjective effort ratings in the full sample, a small positive correlation was found ($r=0.15$, $p=.017$). When studying this correlation for each age group separately, no significant correlations were found (all $p \geq .053$), so this correlation seems to be independent of age group. We did not include these subjective effort scores in the manuscript for conciseness, but we refer other researcher to the openly shared data on OSF to further explore these data (cf. comment 10 of Reviewer 3 as well).

4. Finally, to measure proactive and reactive control in the context of the flanker task, the proportion of incongruent trials was manipulated across three blocks, whose order of presentation was counterbalanced across participants. However, it cannot be excluded that facing first the block inducing proactive (vs reactive) control influences the kind of control implemented in subsequent blocks. Did the authors check if implementation of proactive / reactive control vary according to the proportion of incongruent trial in the first block ?

To check for this, we reran all the analyses for the Flanker task, adding block order to the linear mixed models as fixed effect and built up the random effect structure in a stepwise fashion. For error rates, the analyses failed to converge. For RTs, the model with the best fit included Block order, Block, Congruency and Group and their interactions as fixed effects, and random intercepts for Participant and random slopes for Block. Here again, the more complex models did not converge. No main effect of Block Order was observed, but significant interactions between Block Order and Block, and between Block Order, Block and Congruency were observed. Post-hoc tests for the three-way interaction showed a significantly smaller PCE (i.e. smaller adaptation and thus decreased proactive control exertion) for the block order 87-67-47 ($M=41\text{ms}$) compared to the block orders 47-67-87 ($M=55\text{ms}$; $p=.013$), 67-47-87 ($M=56\text{ms}$; $p=.0055$), and 87-47-67 ($M=56\text{ms}$; $p=.0037$). It thus seems that the cognitive control settings in the first block may impact subsequent blocks, specifically when proportion congruency declines gradually.

However and crucially, no differences in this interaction between groups (i.e. the four-way interaction) were observed ($p=.35$), nor any other interactions including both Block Order and Age Group (all $p \geq .49$). Moreover, when adding the factor Block Order to the model (leading to a different random structure as the original model without block order), the pattern of results regarding age differences, as reported in the manuscript, remained highly similar. The only observed difference relates to the three-way interaction between Block, Congruency and Age group. Post-hoc tests with Tukey multiple comparison correction now showed that the PCEs no longer differed significantly between age groups when block order was included in the model, except for a significantly larger PCE in older adults aged 70-79 ($M=58\text{ms}$) compared to older adults aged 60-69 ($M=48\text{ms}$, $p=.0020$). However, this interaction was already ambiguous in the main analyses, as we discussed in the Discussion as well on p. 17.

We have added a footnote in the Results section indicating that the order of the blocks in this task can have an impact on the size of the observed proportion congruency effects, although the PCEs remained significant regardless of block order (all $p < .001$), and the block order effects did not interact with age.

Results, p. 13: *“Adding the order of the blocks to the models showed that the order of the blocks in this task can have an impact on the size of the observed PCEs. However, the PCEs remained significant regardless of block order (all $p < .001$), and the block order effects did not interact with age. For conciseness, we therefore only included the analyses without block order.”*

References (additional to those in the manuscript):

- Rey-Mermet, A., Gade, M. & Oberauer, K. Should we stop thinking about inhibition? Searching for individual and age differences in inhibition ability. *Journal of Experimental Psychology: Learning, Memory, and Cognition* **44**, 501–526 (2018).
- Friedman, N. P., & Miyake, A. (2004). The Relations Among Inhibition and Interference Control Functions: A Latent-Variable Analysis. *Journal of Experimental Psychology: General*, *133*(1), 101–135. <https://doi.org/10.1037/0096-3445.133.1.101>
- Long, J., Song, X., Wang, Y., Wang, C., Huang, R., & Zhang, R. (2022). Distinct neural activation patterns of age in subcomponents of inhibitory control: A fMRI meta-analysis. *Frontiers in Aging Neuroscience*, *14*. <https://doi.org/10.3389/fnagi.2022.938789>
- Khng, K. H., & Lee, K. (2014). The Relationship between Stroop and Stop-Signal Measures of Inhibition in Adolescents: Influences from Variations in Context and Measure Estimation. *PLoS ONE*, *9*(7), e101356. <https://doi.org/10.1371/journal.pone.0101356>

Violato, E. M., Voorrips, E. S., Desender, K., & Van den Bussche, E. (2025). Metacognitive awareness of subjective difficulty, effort, and frustration in cognitive conflict contexts. *Motivation Science*. Advance online publication. <https://doi.org/10.1037/mot0000392>

Reviewer #3:

The authors present a manuscript in which they examine changes of subcomponents of cognitive control at different ages. This is an interesting and timely study with a relatively large number of participants. The task selection to analyze cognitive control, the methods, data analysis and result presentation sound well. However, there are some missing information, methodological problems and problems with the data analysis that may lead to incorrect conclusions. Overall, there is some space for improvement. I will go through the manuscript and comment on the problems in the order in which they occur.

Ref:

Hultsch, D.F., MacDonald, S.W.S., 2004. Intraindividual variability in performance as a theoretical window onto cognitive aging. In: Dixon, R.A., et al. (Eds.), *New Frontiers in Cognitive Aging*. Oxford University Press, Oxford, pp. 65–88.

MacDonald, S.W.S., Nyberg, L., Bäckman, L., 2006. Intra-individual variability in behavior: links to brain structure, neurotransmission, and neuronal activity. *Trends Neurosci.* 29, 474–480.

Ratcliff, R., 1978. A theory of memory retrieval. *Psychol. Rev.* 85, 59–108.

Townsend, J.T., Ashby, F.G., 1983. *Stochastic Modeling of Elementary Psychological Processes*. Cambridge University Press, Cambridge.

1. Overall, the authors used 4 groups of participants with young 18-27 years old and older beginning with the age of 60. That is, there is a large age gap in the middle-age between 27 and 60 years preventing conclusions about cognitive changes across the adult lifespan. It is well known that cognitive changes begin in the early fifties due to progressive thinning of the cortex, mainly at prefrontal areas responsible for executive and cognitive control functions.

We agree that this study lacks an intermediate group, to be able to detect changes over the whole life span. This is discussed in the Discussion section.

Discussion, p. 18: "this study did not include a middle age group (28-59), which keeps us from being able to show the full adult lifespan changes. For this reason, we chose not to treat age as a continuous variable, but rather use separate age groups. Future studies including this age group would be useful."

2. The difference in education between age groups is surprising. Young adults had a lower number of years than the older ones or even the oldest group. In most European countries, the education level after the second world war (80 or 70 years ago) was only rudimentary and mostly elementary school was finished. Please explain this discrepancy in more detail and include the range of education years in the table. Secondly, the ratio of females was larger in the young than the older groups that as well as higher MoCA scores in the young-old than old-old group also needs some explanation. Please compare statistically the groups regarding GDS-15 and years of education.

We compared GDS scores and education statistically between age groups and added this to the Results section.

Results, p. 6-7: *“Moreover, differences in scores on the GDS-15 ($F(3,302)=7.51, p<.001$) and in education ($F(3,301)=4.14, p=.0068$) were observed between groups. Young adults had significantly higher GDS-15 scores than the older adults ($p\leq.0067$). In addition, older adults aged 60-69 reported having a higher education compared to the other age groups ($p\leq.042$).”*

In addition, ranges of education were added to Table 2. Please note that young adults in this study were first year students at the university, thus they had not completed their education yet at the time of the study. Comparing education between young and older adults is thus not very meaningful here. This was also added to the Results section in the manuscript.

Results, p. 7: *“Note that comparisons with the young adults group are not very meaningful, as this group’s education was still ongoing.”*

Regarding education in older adults, indeed, lower education is expected with older age (for example, for Belgium see: <https://statbel.fgov.be/en/themes/census/education/level-education>). As indicated in the discussion, as the older adults were highly educated in this sample, we should remain cautious when generalizing these results to the larger population.

Discussion, p. 19: *“However, older adults were required to be cognitively healthy (cf. MoCA and GDS inclusion criteria and self-reported diseases), were mainly highly educated, and living independently at home, which could have led to an underestimation of cognitive control declines with age.”*

For the GDS-15 differences between groups, we refer the reviewer as well to comment 3 of Reviewer 2. We excluded participants already based on GDS-15 scores, so these differences are sub-clinical differences. Moreover, when excluding participants with a score above 4 on this questionnaire (mainly young adults), results remained highly similar and GDS-15 differences between groups were no longer significant.

We agree that the larger ratio of females in young than older adults is a limitation of this study and have mentioned this in the discussion.

Discussion, p. 18-19: *“Moreover, the young adults included in this study were mainly undergraduates with a mean age of 18 and predominantly female. A more balanced sample should be included in future studies.”*

Lower MoCA scores with age are to be expected, as cognitive functioning is expected to decline with age. For example, earlier research that provides MoCA norm scores in older adults over 70 years old already showed lower MoCA scores with older age (Malek-Ahmadi et al., 2015).

3. The authors obtained detailed sociodemographic questions which is positive. Is there any information about marital status, loneliness, quality of life, regular physical or cognitive activity, nutrition etc., i.e., factors that are important for cognitive functioning in older age? From the supplementary table 2 it was surprising to see that there were only a few diseases in the older groups. Depression symptoms, diabetes, heart or cardiovascular disease are common in older population but not in the present sample. Do the authors have an explanation for this or were they a priori excluded from the sample? The most important question is how representative is the sample compared to a general population. It would be helpful if the authors could provide some reference parameters to assess the representativity.

We assessed no other demographic information than the one reported in the supplementary material. Note that having a (history of) psychiatric illness or neurodegenerative diseases, having had cancer treatment in the last two years, and drugs or alcohol abuse (all self-reported) were used as exclusion criteria to ensure that we were including older adults who were cognitively

healthy (see Methods, p. 21), which can explain the lower prevalence of certain diseases. Furthermore, to be included in the study, participants already had to score 6 or lower on the GDS-15, a questionnaire assessing depressive symptoms. So indeed, our aim to include cognitively healthy participants very likely made our sample less representative for the general population. However, as diseases or disorders potentially impacting cognitive health would introduce significant biases to our design, we opted for the current recruitment strategy. Hence, as in most aging research, a selection bias is probably present. Indeed, a health survey from Sciensano in Belgium in 2023 (Van der Heyden et al., 2025) showed that 40% of the older adults aged 65 years or older reported having a chronic disease. For example, around 15% of older adults reported having diabetes and about 8.6% reported having severe heart diseases (averaged over sex), compared to 0.87% for both diseases in our study. In addition, in Belgium almost 6% of older adults aged 65 years or older reported having suffered from a depression the past year (Van der Heyden et al., 2025). Moreover, the demographic information in our study was based on self-report, and the related question only discussed any diseases that could have an influence on cognitive functioning. Hence, not all diseases may have been reported by the participants. This bias is discussed in the Discussion.

Discussion, p. 19: *“In addition, as in most aging research, participants motivated to enroll in the study could already have been cognitively stronger, inducing a selection bias. We tried to keep the inclusion criteria as inclusive as possible, and conducted the participant recruitment very extensively and intensively. However, older adults were required to be cognitively healthy (cf. MoCA and GDS inclusion criteria and self-reported diseases), were mainly highly educated and living independently at home, which could have led to an underestimation of cognitive control declines with age. Indeed, a health survey from Sciensano in Belgium in 2023 (Van der Heyden et al., 2025) showed that 40% of the older adults aged 65 years or older reported having a chronic disease. For example, around 15% of older adults reported having diabetes and about 8.6% reported having severe heart diseases (averaged over sex). This is in clear contrast to the reported 0.87% for both diseases in our study. In addition, Van der Heyden et al. (2025) reported that almost 6% of adults aged 65 years or older reported having suffered from a depression during the past year. Again, this is not in line with our sample where a score of six or lower on the GDS-15 was used as an inclusion criterion. Moreover, the demographic information in our study was based on self-report, and the relevant question pertaining to diseases only mentioned diseases that could have an influence on cognitive functioning. Hence, not all diseases may have been reported by the participants.”*

4. Methods: cognitive test battery. The tasks and questionnaires were applied at home. Where there an experimenter who accompanied the session, or did the participants conduct the test battery self-paced? If so, how did the authors control compliance with the test program and control distraction and other individual parameters? Are different laptops / PC used for data collection? This is important for the comparability of data sets. Please discuss in the section Limitations of the study.

One experimenter was always present during data collection, to explain the task to the participants and aid during practice trials. After that, once the experimental trials started, the participant completed the task by themselves. Different laptops were used, but always with an identical keyboard and mouse. Moreover, by using visual degrees in the programming of the task and setting the screen width correctly for each laptop that was used, stimuli were always presented in the same size, regardless of the type of laptop that was used for data

collection, making the collected data highly comparable. This is clarified and discussed in the Methods section.

Methods, p. 22: *“In addition, the screen width of the laptop on which the tasks were administered was entered, to make sure that stimulus sizes were identical for all participants, independent of the laptop used for data collection.”*

Methods, p. 27: *“Participants were tested at home, keeping in mind the COVID-19 regulations that were in place during the data collection of this study (June 2021 until January 2023). An experimenter was present during data collection.”; “The experimenter first explained the task and monitored practice trials, whereafter participants completed the task themselves.”; “An external keyboard and mouse were used in addition to the laptop, to administer the tasks.”*

5. It seems that in the Go/NoGo task there was no time pressure to avoid waiting strategy. The core of the task is to press a response button as fast as possible to provoke failures of inhibitory control (i.e. to produce no-go errors). The problem with this task is that the same response deadline of 500ms was applied in all age groups and it is known that older participants show longer RTs than younger ones (age-related slowing). A much better solution would be an adaptive deadline individually computed for each participant based on mean RTs (or mean + 1SD). This would ensure comparable time pressure in young and in old participants. The supposed shift of strategy between young and old could be due to different feelings of time pressure. This methodical shortcoming having generally huge effects on the cognitive performance and conclusion about age-related differences in inhibitory control should be discussed in the section Limitations of the study.

Older adults indeed were slower on this task compared to young adults. However, mean reaction times were still well below the 500ms response deadline ($M_{60-69}=351\text{ms}$, $M_{70-79}=362\text{ms}$, $M_{80+}=375\text{ms}$). Moreover, none of the older adults had to be excluded on this task for reaching less than 25% accuracy. This indicates that older adults were able to perform the task with the set response deadline. In addition, when looking in more detail at the reported effort on the NASA-TLX questionnaire (not included in the manuscript for conciseness), young adults reported significantly more effort for the Go/No-Go task than the 70-79 group, in line with the behavioral results when focusing on commission errors.

We agree however with the reviewer that an adaptive response deadline would be an interesting approach in future research to make the experienced time pressure comparable across participants and age groups. We have added this to the Discussion.

Discussion, p. 19 *“Finally, for tasks with strict response deadlines, such as the Go/No-Go and N-Back, the same response deadline was used across all age groups. However, an adaptive response deadline could be an interesting approach in future research, to make the experienced time pressure more comparable across participants and age groups.”*

6. The authors used RTs and error rates as dependent variables. However, in cognitive tasks comparing different age groups speed-accuracy tradeoff may occur. This is apparent for example in the Go/NoGo tasks, where the young group show the highest error rates and fastest responses. On the other hand, there is age-related slowing, as well lower error rates in older age, mainly in difficult e.g., incongruent or switch conditions. How can the authors exclude the speed-accuracy tradeoff? There are some ways to solve the problem and to weight both parameters equally. For example, by computing inverse efficiency scores (IES; Townsend and Ashby, 1983). Alternatively, there is the possibility to compute the drift rate

obtained using a drift diffusion model (Ratcliff, 1978). Both parameters, IES and drift rate, reflect a compound of speed and accuracy and would make it possible to compare overall performance between age groups.

We agree with the reviewer that speed-accuracy trade-offs could have occurred in the tasks. We have put a lot of thought and effort in this comment to try out and decide what the best approach here would be. This decision is complicated by the fact that the tasks differ in design and measured outcome variables (e.g., RT, error rates, completion times, etc.), so it is difficult to find a single good solution for all tasks. Furthermore, several approaches to account for speed-accuracy trade-off (e.g., calculating the Inverse Efficiency Score or IES), require aggregating the data again, which would counteract all the advantages of using (G)LLMs, where trial-level instead of aggregated data are analyzed. Furthermore, using IES (where RT is corrected by dividing by accuracy) for tasks such as the Go/No-Go are tricky, as the main outcome variable there is No-Go accuracy, rather than Go RTs. However, given the strict response deadline for this task, speed-accuracy trade-offs might also be less likely here. In addition, the use of IES has been debated as well in the literature, especially in tasks with error-rates of 10% or higher (Bruyer & Brysbaert, 2011), and this compound measure still seems to be sensitive for speed-accuracy trade-offs (Liesefeld & Janczyk).

Ultimately, we opted to address this speed-accuracy trade off by including and thereby controlling for reaction time (or completion time for the Plus-Minus task) into the generalized linear model with error rate as dependent measure for each of the tasks (e.g. in line with Davidson & Martin, 2013). To limit complexity and increase chances of model convergence, a variable containing mean RT per participant for each task was calculated and this variable was then added as a fixed main effect in the models. This means that we controlled for the *overall* RT in a given task. This also allowed us to perform this analysis for the Go/No-Go and N-Back tasks, where No Go and non-target performance are typically not accompanied by an RT (i.e., these trials require withholding a response, so an RT is only recorded on the few occasions when a response is erroneously provided). By using overall RT across the task, we can still detect if RT strategies impacted error rates (e.g., do participants with a slower overall RT on a given task also show lower error rates?), and whether such a speed-accuracy trade-off impacted our observed pattern of results. Please note that these analyses are very time-consuming to run and add a lot of complexity to the paper.

When controlling for RT in the models in the way described above, error rate analyses showed a highly similar pattern of results compared to the original analyses. The observed differences in cognitive control subcomponents found between age groups thus still remain even after controlling for speed-accuracy trade-off. A detailed description can be found in the Supplementary Results and we refer to it in a footnote in the Results section.

Results, p. 7: *“When controlling for overall RT, by calculating the mean RT across all trials for each participant and each task and adding this variable as a fixed main effect in the error rate models, we observed a highly similar pattern of results, with all age group effects still present for all tasks. A detailed description of these analyses can be found in the Supplementary Results.”*

7. How long was the response deadline in the Flanker and the AX-CPT tasks? Overall, in both tasks applied in different age groups there is the same problem with speed deadlines, as mentioned above.

No response deadlines were applied in these tasks (only the instructions stress speed and accuracy), so we don't expect differences related to speed deadlines between age groups for

these tasks. We refer the reviewer to our reply to comment 6 above where we took into account the speed-accuracy tradeoff in our analyses, also for these tasks.

8. *It is unclear what does mean the 30% hit rate in the N-back task. Is it the rate of targets? Please provide the number for targets and non-targets for each task. Please rename the error types in the common ones (omissions or missed targets) and false alarms (response to non-target).*

We thank the reviewer for this comment. We have renamed the terminology for the N-Back task throughout the manuscript. For example (in bold):

Methods, p. 24-25: *“Participants had to decide for each stimulus whether the presented picture was identical to the picture that was presented second-to-last (i.e., 2-back task), by pressing “J” as quickly as possible if that was the case (i.e. **target trials**). All used pictures were validated in a previous study of Rossion and Pourtois. Participants received one practice block of 50 trials and two main blocks of 50 trials with a break in between, both with a **30% rate of target trials (i.e. 15 target trials and 35 non-target trials per block)**. The dependent variables were RTs and errors at the trial level. This enabled us to analyze mean RTs on the target trials, the percentage of errors on target trials (when the participant should have pressed the key but did not, i.e. **omission errors or missed targets**), and the percentage of errors on non-target trials (when the participant pressed the key but should not have, i.e. **commission errors or false alarms**). **Omission errors or missed targets** are often used as the main index of updating in this task.”*

9. *In the task switching, were the single task blocks averaged and then subtracted from the switch block? Were the trial by trial RTs recorded and analyzed or the total time to perform a block? This is unclear. In general, there is a lack of theoretical background for the task switching. Moreover, it is unclear why both global (mixing) and local switch costs were not reported. This specific design only became clear during the discussion and needs to be introduced earlier.*

We have clarified this in the Methods section of the manuscript.

Methods, p. 25: *“In the Plus-Minus task, measuring shifting, a list of random two-digit numbers was presented. In a first block, participants had to add 3 to each two-digit number (ranging from 13 to 96), in a second block they had to subtract 3 from each number, and in a last block they had to alternate between adding 3 and subtracting 3. Participants received a list of 6 numbers as a practice list for each block. In each main block, participants completed a list of 30 numbers (size=40), **presented on a single screen**. Feedback was presented after completing a list. **For each list, the completion time of the full list** and total number of errors were measured. **The dependent variables were the required time to complete a list (in seconds) and the total number of errors for each list, which enabled us to compare these outcomes between the non-switch and the switch blocks at the participant level. A switch cost was calculated as the difference between the completion time or errors on the switch block and the average completion time or errors on the no switch blocks, i.e., the global switch cost as an index of shifting. Due to the design of the Plus-Minus task, with only total time to complete a list and total accuracy per list, we were not able to compute a local switch cost.”***

10. *Were the results of the digit span correlated with the cognitive tasks?*

Both the forward and backward digit span correlated negatively with missed targets (i.e. omission errors) on the N-Back task and RT on AY trials of the AX-CPT (i.e. larger span, better updating and faster on AY trials).

As the digit span was outside the scope of this manuscript we have not added this correlation table to the manuscript, to keep the manuscript focused and concise. However, as all data are available on OSF, other researchers can further use and study these data. We have now explicitly mentioned this in the Methods section.

Methods p. 28: *“The cognitive motor-dual task, basic Reaction Time task, NASA-TLX scales, LSNS-6 and WAIS-IV digit span go beyond the scope of this study and won’t be discussed here. We refer other researchers to the OSF page (<https://osf.io/vbh58/>) to further explore these data, only excluding those data that are still embargoed as they are a part of a different study (i.e., cognitive motor-dual task).”*

11. Inhibition was studied using 47C block of the Flanker task. Why? Thus, the overall congruency effect was rather small (20 ms). Inhibitory control is mostly required in blocks with rare targets (87C) and the congruency effect should be much larger. The small congruency effect (together with large data variability in older age, particularly in difficult conditions) could be the reason for no significant difference between the young and the 80+ group. This may lead to the wrong conclusion that older groups show superior cognitive control compared to the younger group. Additionally, the authors interpreted the 47C data and suggested that more extreme differences in proportion congruency conditions are needed. This procedure is unclear.

To study inhibition, we selected the block with the most “neutral” proportion congruency, where conflict is unpredictable and no expectations could be formed based on congruency, as both congruent and incongruent trials occurred almost equally. A 87C block indeed triggers large congruency effects, given the rare incongruent trials. We reran the analyses only on this 87C block as a sanity check, instead of the 47C block. Results were highly similar. The overall congruency effect for the 87C block was indeed larger ($M=70\text{ms}$) compared to the 47C block ($M=20\text{ms}$). Similarly as for the analyses of the 47C block, the interaction between Congruency and Age group for reaction times in the 87C block was significant ($F(3,295.86)=13.73, p<.001$). Post-hoc tests with Tukey correction for multiple comparisons showed that the CE was significantly larger in young adults ($M=70\text{ms}$) compared to older adults aged 70-79 ($M=65\text{ms}, p=.046$). No differences in CE were found between young adults and the 60-69 and 80+ group, nor within the older age cohorts. This is similar to what we observed for the 47C block. Hence, the same pattern of results also emerges when only taking into account the 87C block that triggers the largest congruency effects.

Our suggestion to use more extreme differences in proportion congruency concerned the analyses regarding proactive and reactive control, in which we took into account all three blocks of the Flanker task, with proportion congruency manipulation (i.e. 87%, 67% and 47% congruency):

Discussion, p. 17: *“Results for the Flanker task with proportion congruency manipulation showed larger congruency effects in young compared to older adults. This is in line with.... Results for proactive control, indexed by the Proportion Congruency Effect were unclear. Future research is needed, with a stronger manipulation of proportion congruency (e.g., more extreme differences in proportion congruency conditions), to study age-related differences in proactive control.”*

See also the Methods section p. 24, where we clarified which blocks were included in which analyses for the Flanker task: *“The 47C block was used to study interference control, whereas all blocks were included in the analyses of proactive and reactive control.”*

12. In Fig 1. The Flanker congruency effect was measured in milliseconds and in %. But there are some negative values. Specifically, almost all values in the age group 70-79 are negative. Why?

Negative values indicate faster reaction times or fewer errors on incongruent compared to congruent trials, i.e. a reverse congruency effect. This is not uncommon, as congruency effects will fluctuate within and between participants. However, this (reversed) difference between congruent and incongruent trials in errors was not significant for the age groups 70-79 and 80+ ($p \geq .081$).

13. Are the results in the section *“Flanker with proportion congruency manipulation”* across all 3 blocks? As the interaction *Age x Block x Congruency* was significant, I would suggest reporting the results of the *Age x Congruency* interaction for each block separately. In other words, the CE should be compared between the age groups for each block separately. This allows to see how the CE changes across the age depending on the ratio of incongruent trials and would increase the transparency.

Indeed, as clarified in the Method section on p. 24, to study proactive and reactive control we used the Flanker task with all three blocks: *“The 47C block was used to study interference control, whereas all blocks were included in the analyses of proactive and reactive control.”*

As the main index of interest here for this interaction is the PCE and how this differs between age groups, to study proactive control differences, we opted to include this in the main section of the manuscript. However, studying how the CE changes with age depending on the block could also be interesting. We have added the requested analyses to the supplementary materials (see Supplementary Results).

Regarding reaction times, for the 87C block, the CE did not differ significantly between age groups ($p = .051$). For the 67C block, the CE was significantly larger in the 80+ group ($M = 49\text{ms}$) compared to the other age groups ($M_{\text{young adults}} = 37\text{ms}$, $M_{60-69} = 36\text{ms}$, $M_{70-79} = 35\text{ms}$; all $p < .001$). Finally, for the 47C block, the CE was larger in young adults ($M = 23\text{ms}$) compared to the 60-69 ($M = 12\text{ms}$, $p < .001$) and 70-79 group ($M = 17\text{ms}$, $p = .030$). Moreover, the CE was significantly larger in the 80+ group ($M = 21\text{ms}$) compared to the 60-69 group ($M = 12\text{ms}$, $p < .001$).

Regarding error rates, for the 87C block, the CE was significantly larger in young adults ($M = 2.88\%$) compared to the older age groups ($M_{60-69} = 0.27\%$, $M_{70-79} = 1.32\%$, $M_{80+} = 1.40\%$; all $p < .001$). Moreover, the CE was significantly smaller in the 60-69 group compared to the oldest age cohorts ($p \leq .0012$). For the 67C block, the CE did not differ significantly between age groups ($p = .83$). Finally, for the 47C block, the CE was significantly larger in young adults ($M = 1.64\%$) compared to the older age groups ($M_{60-69} = 0.57\%$, $M_{70-79} = -0.46\%$, $M_{80+} = 0.11\%$; all $p < .001$). Moreover, the CE was significantly smaller in the 70-69 group compared to the 60-69 and 80+ groups ($p \leq .042$).

In general, the CE was larger in young adults compared to older adults, especially for the 47C block. Moreover, the CE in the 80+ group was larger than the other age groups for most blocks.

14. When looking at the mean RTs in Figures 1 and 2 there is an apparent increase of RT variability with age. My suggestion is to use an appropriate measure and to analyze the variability as an important parameter of cognitive performance in age that allow conclusions about functional stability or “noise” in the brain. (c.f. Hultsch & MacDonald, 2004; MacDonald et al., 2006). This seems to be important for the present study as acknowledged in the Conclusions. Additionally, due to the age-related slowing and high RT-variability the difference values such as CE are less valid or show small or even negative values (longer RTs in congruent than incongruent trials).

For the analyses, we used (generalized) linear mixed models. This type of analysis already accounts for intra-individual variability by incorporating random intercepts for participants and slopes for conditions, based on the best model fit (Baayen et al., 2008). Moreover, using linear mixed models, we analyzed the data at the trial level, rather than aggregating the data and losing within-participant variability and reducing power (Judd et al., 2012). Only for post-hoc tests and ease of interpretation, we aggregated the data (based on predicted means from the models) into congruency effects, for example, to further interpret the found interactions.

For the CE's, this participant variability has thus been taken into account. Post-hoc tests showed that there was no significant difference between the incongruent and congruent trials in the two oldest age groups, see comment 12 as well.

15. It is unclear why X^2 test was used instead a F -test. Why did the authors use binomial error distribution instead of individual mean error rates? Is box-cox transformation similar to Log10 transformation to transform continuous data to normal distribution? Did the authors prove that the data is indeed normally distributed after the transformation?

Using (generalized) linear mixed models, data are analyzed at the trial level instead of being aggregated to means, as in ANOVA (see also the previous comment). For the analyses with error as the outcome variable, this means that on each trial, the error variable can have either a value of 0 (correct trial) or 1 (incorrect trial), thus being a binary outcome. This is typically modeled in a generalized linear mixed model with a binomial distribution.

See the Methods section as well, p. 28: “For the analyses of the errors, GLMMs were applied to the binary data (0=correct and 1=incorrect) using a binomial error distribution.”

To assess the significance of the main and interaction effects in a generalized linear mixed model, we employed a Wald test, a method commonly used in the field (e.g. Arango-Botero et al., 2023; Fromont et al., 2020). This test yields a X^2 statistic, as opposed to the Satterthwaite approximation method we used in the linear mixed models in the manuscript, which yields an F -test.

We refer the reviewer to the Methods section, p. 28: “The significance of the main and interaction effects was assessed with a Type III ANOVA with the Satterthwaite approximation method for the LMMs and the Wald test for the GLMMs.”

For reaction time analyses, we used linear mixed models. Violations of assumptions were studied visually (see also the openly shared code on OSF). To accommodate for non-normality and heteroscedasticity, a Box-Cox transformation was applied to the reaction times for the Go/No-Go and Flanker tasks. The Box-Cox transformation is a family of power transformations that includes the log transformation as a special case (when the Box-Cox parameter $\lambda = 0$). While a Log10 transformation is a fixed transformation, the Box-Cox method estimates the optimal λ to best approximate normality and stabilize variance.

Therefore, Box-Cox is more flexible and can be more effective in normalizing data. We refer the reviewer to Atkinson et al. (2021) for a review, and have added this reference to the manuscript as well.

Methods, p.28: “When assumptions were violated (i.e., for the Go/No-Go and Flanker tasks), Box-Cox transformations on the RTs were applied (for a review, see Atkinson et al., 2021).”

After Box-Cox transformation, assumptions were not violated anymore based on visual inspection. In addition, we have also ran generalized linear mixed models for the RT analyses, to account for any non-normal data. As results remained highly similar, we opted to include the more easily interpretable linear mixed models using Box-Cox transformations in the manuscript.

References (additional to those in the manuscript):

- Malek-Ahmadi, M., Powell, J. J., Belden, C. M., O'Connor, K., Evans, L., Coon, D. W., & Nieri, W. (2015). Age- and education-adjusted normative data for the Montreal Cognitive Assessment (MoCA) in older adults age 70–99. *Aging, Neuropsychology, and Cognition*, 22(6), 755–761. <https://doi.org/10.1080/13825585.2015.1041449>
- Bruyer, R., & Brysbaert, M. (2011). Combining Speed and Accuracy in Cognitive Psychology: Is the Inverse Efficiency Score (IES) a Better Dependent Variable than the Mean Reaction Time (RT) and the Percentage Of Errors (PE)?. *Psychologica Belgica*, 51(1), 5-13. <https://doi.org/10.5334/pb-51-1-5>
- Liesefeld, H. R., & Janczyk, M. (2019). Combining speed and accuracy to control for speed-accuracy trade-offs (?). *Behavior Research Methods*, 51, 40-60. <https://doi.org/10.3758/s13428-018-1076-x>
- Davidson, D. J., & Martin, A. E. (2013). Modeling accuracy as a function of response time with the generalized linear mixed effects model. *Acta psychologica*, 144(1), 83-96. <https://doi.org/10.1016/j.actpsy.2013.04.016>
- Baayen, R. H., Davidson, D. J., & Bates, D. M. (2008). Mixed-effects modeling with crossed random effects for subjects and items. *Journal of memory and language*, 59(4), 390-412. <https://doi.org/10.1016/j.jml.2007.12.005>
- Judd, C. M., Westfall, J., & Kenny, D. A. (2012). Treating stimuli as a random factor in social psychology: a new and comprehensive solution to a pervasive but largely ignored problem. *Journal of personality and social psychology*, 103(1), 54. <https://doi.org/10.1037/a0028347>
- Arango-Botero, D., Hernández-Barajas, F., & Valencia-Arias, A. (2023). Misspecification in Generalized Linear Mixed Models and Its Impact on the Statistical Wald Test. *Applied Sciences*, 13(2), 977. <https://doi.org/10.3390/app13020977>
- Fromont, L. A., Steinhauer, K., & Royle, P. (2020). Verbing nouns and nouning verbs: Using a balanced design provides ERP evidence against “syntax-first” approaches to sentence processing. *PLoS One*, 15(3), e0229169. <https://doi.org/10.1371/journal.pone.0229169>

Manuscript COMMSMED-25-0075-A "Not All is Bad with Older Age: a Comprehensive Overview of Cognitive Control in Healthy Aging"

Reviewers' comments:

Reviewer #1 (Remarks to the Author):

The Authors have addressed my concerns. I have no further comment.

Reviewer #2 (Remarks to the Author):

The authors have responded satisfactorily to my comments and requests for clarification (apologies for my sentence with typo about your results in my previous report).

Reviewer #3 (Remarks to the Author):

1. The reviewer appreciates the effort the authors invested in the revision of the manuscript. The authors responded adequately to almost all my comments and discussed the problems with the demographic data and representativeness of the sample, but the section should be named "Limitations" instead of "recommendation for future research".

Response: Thank you for this suggestion, we have adapted the title of this section to "limitations and recommendations for future research". This section not only discusses limitations in the current study, but also provides guidelines and critical points to address in future aging research.

Discussion, p. 31: "Regarding limitations and recommendations for future research,".

2. However, in my opinion the issue with the control of the speed-accuracy tradeoff is not completely solved and the reviewer is still not fully convinced. This is an important problem as the design of the study and the data analysis strategy may considerably impact the conclusions. Regarding Go/NoGo task the strategy to perform the task in young and older adults seems to be different. Nevertheless, the conclusion of the findings was that there is no inhibitory decline on older age. This may lead to wrong conclusions such as "This pattern speaks against a general decline in inhibition as found by other cross-sectional studies 11,22–25 and general deficit theories such as the Inhibition Deficit theory" which is simply not true. This is a strong conclusion and is rather speculative as the data were not analyzed in the most common way using, for example mixed ANOVAs with aggregated data. The additional results can be included in the supplementary material.

Response: We respectfully disagree with the reviewer's statement of not analyzing the data in the most common way and thus making speculative conclusions. In many domains of psychology and neuroscience, researchers are urged to move towards using linear mixed effects modeling instead of analyses based on aggregated data such as (mixed) ANOVAs, especially when data are hierarchical or correlated such as the repeated measures in the current study (Judd et al., 2012; Yu et al., 2022). Applying linear mixed models, data are analyzed at the trial level instead of being aggregated to means as in ANOVA. This means that all data are used in the analyses. Hence, in this way, we avoid losing within-participant variability and reducing power as is the case in ANOVA (Judd et al., 2012). In addition, by incorporating random intercepts for participants and random slopes for conditions, intra-individual variability is incorporated and controlled for in the models as well. Especially when having a repeated measures design, as is the case in most of the tasks here, linear mixed models are crucial to correctly analyze the data and take into account the data dependence, which is typically

ignored when using ANOVA (Meteyard & Davies, 2020; Yu et al., 2022). Moreover, Communications Medicine has also published papers using linear mixed models before (e.g. Dashti et al., 2025; Tang et al., 2025).

Nevertheless, we have reran the analyses for the Go/No-Go task using ANOVAs with aggregated data. We obtain the same results as with the linear mixed models: commission errors significantly decreased with age ($F(3,302)=3.56, p=.015$), with young adults making more commission errors ($M=14.10\%$) compared to older adults ($M_{60-69}=10.34\%$, $M_{70-79}=9.05\%$, $M_{80+}=11.74\%$), although the only significant difference after correction for multiple comparisons was the one between young adults and the 70-79 group ($p=.004$), identical to the output of the generalized linear mixed model in the manuscript. As linear mixed models provide a more robust analysis, we opted to include these in the manuscript and not these ANOVAs. However, we provide the full dataset openly online, according to the open access principles. We invite other researchers to further explore the data, for example to use mixed ANOVAs with aggregated data.

We have put a lot of thought and effort into coming up with the optimal analytical approach to handle the speed-accuracy trade-off in the previous revision. Ultimately, we have addressed this trade-off, by controlling for reaction time in the error analyses, in line with Davidson and Martin (2013). This way, we could detect if RT strategies impacted error rate performance and if age-related patterns still remained after controlling for this (which they did). However, we agree that we can also include other measures such as the IES, which we address in the next comment.

3. I think this was a misunderstanding about computing the IE Score. IES reflects the relationship between the mean individual speed in correct Go trials and the ratio of commission errors. The simplest way to test speed-accuracy trade-off is to correlate RTs and Error rates across all participants or separately for each age group. Nevertheless, I encourage the authors to compute the individual inverse efficiency score integrating both behavioral parameters and to add this to the manuscript as a simple but informative index of performance:

*The computation for the values Go/NoGo task of the averaged values is: Mean RT in Go trials / (100 – mean ratio of false alarms in NoGo trials) * 100*

*For the 2-back task: Mean RT in correct target / (100 – mean ratio of missed targets) * 100*

*For the AX-CPT (either across all conditions or separately for each condition): Mean RT in correct trials / (100 – mean ratio of errors) * 100*

*Finally, in the plus minus task, the authors may provide individual RT in the simple blocks and the switch block / (100 – mean ratio of errors in the corresponding block) * 100*

Response: Thank you for these clarifications. We have now calculated the IES for each task as suggested above, across conditions, and have included these measures in the Supplementary materials as well (see Supplementary Results, Addressing speed-accuracy trade-offs: Comparing the Inverse Efficiency Scores, p. 15-17). Results remain consistent to the original RT analyses included in the main manuscript. To summarize, even when controlling for accuracy and thus accounting for strategy differences, age-related slowing was still observed. A detailed description can be found in the Supplementary Results and we refer to it in a footnote in the Results section.

Results, p. 19, footnote 4: “When controlling for overall RT, by calculating the mean RT across all trials for each participant and each task and adding this variable as a fixed main effect in the error rate models, we observed a highly similar pattern of results, with all age group effects still present for all tasks. In addition, comparison of Inverse Efficiency Scores between groups resulted in highly similar patterns as the RT analyses without controlling for accuracy. A detailed description of these speed-accuracy trade-off analyses can be found in the Supplementary Results.”

As we believe that the IES is still a biased, mainly speed-based measure (Bruyer & Brysbaert, 2011; Liesefeld & Janczyk, 2019), we have included both the generalized linear mixed models for errors controlling for speed, as well as the IES analyses in the Supplementary materials. We have added a part to the Discussion as well, summarizing the results of these additional analyses.

Discussion, p. 27: *“Other studies have also shown more commission errors in young compared to older adults. It has been suggested that, rather than worse inhibition in young adults, this could reflect a more cautious response style in older adults, trading speed for accuracy. However, even when controlling for speed in the error analyses (see Supplementary results: “Addressing speed-accuracy trade-offs”), commission errors and congruency effects based on errors were still larger in young compared to older adults. Still, Inverse Efficiency Scores were larger in older compared to young adults, showing reduced efficiency on the task with age. The latter seems to indicate a case of general slowing rather than reduced inhibition, as this pattern is similar to what we observed in the RT results.”*

Discussion, p. 30: *“General age-related slowing was observed across all tasks. This slowing could reflect difficulties with the underlying subcomponent measured by that task. However, this slowing could also be interpreted in light of older adults generally needing more time to process and store information or to respond to a task, in line with the Processing-Speed theory. Even when controlling for speed-accuracy trade-offs (see Supplementary results: “Addressing speed-accuracy trade-offs”), age-related results remained highly similar, potentially reflecting this general age-related slowing.”*

4. *Regarding switching task: The reviewer has understood that the task only provides one speed value per block. However, the determination of a total performance time contains large error variance. This can be significantly reduced by using multiple trials in each condition. The more trials, the lower the variability and error variance in the data. If the task was presented on a trial-by-trial basis, why were the individual RTs not analyzed since each trial was completed with a button press? Or did the participants give the answers verbally? This is unclear. Please discuss this point in the section “Limitations”*

Response: As specified in the Methods section, participants received a list of 30 numbers on the screen (presented in 3 columns of 10 numbers) in each main block. They had to complete the whole list before clicking the “finish” button, after which feedback was presented. Hence, we only have one single speed and accuracy measure per list. This task was developed based on the existing format of the Plus-Minus task (see Miyake et al., 2000; originally adapted from Jersild, 1927, and Spector & Biederman, 1976), which is often administered on paper, with one list for each block and one measure of speed and accuracy per block. Hence, in line with this, no trial-by-trial analyses were performed in this study.

Methods, p. 12: *“In each main block, participants completed a list of 30 numbers (size=40), presented on a single screen (in three columns of 10 numbers). Participants were asked to type the answer next to each number. Feedback was presented after completing a list. For each list, the completion time of the full list and total number of errors per list were measured. The dependent variables were the required time to complete a list (in seconds) and the total number of errors for each list, which enabled us to compare these outcomes between the non-switch and the switch blocks at the participant level. A switch cost was calculated as the difference between the completion time or errors on the switch block and the average completion time or errors on the no switch blocks, i.e., the global switch cost as an index of shifting. Due to the design of the Plus-Minus task, with only total time to complete a list and total accuracy per list, we were not able to compute a local switch cost.”*

We have added this limitation to the Discussion as well.

Discussion, p. 32: *“In addition, as already discussed above, future studies should administer a shifting task that also allows to study local switch costs, which was not the case in the current Plus-Minus task. Moreover, as only one speed and accuracy index are obtained per list in the classic Plus-Minus task, error variance might have been large here. Providing multiple trials could improve interpretability of age effects for shifting.”*

5. *Negative values in the flanker task (incongruent – congruent) indicate slower responses in congruent vs. incongruent trials in the oldest group. This supports the age-related slowing and increasing intraindividual variability of speed in older age that may blur the differences between congruent and incongruent trials. In my view this is a clear indication for different strategies to perform the task between young and old adults and should be extensively discussed.*

Response: Although negative values were present on average in the oldest groups, this reversed congruency effect was not statistically significant, so we should not interpret this as if it were significant. However, we agree that different strategies may have been used in older compared to young adults, especially for inhibition. This is discussed in the Discussion section of the manuscript.

Discussion, p. 27: *“Other studies have also shown more commission errors in young compared to older adults. It has been suggested that, rather than worse inhibition in young adults, this could reflect a more cautious response style in older adults, trading speed for accuracy.”*

Crucially however, larger congruency effects in young compared to older adults were still observed in the error analyses even after controlling for speed and controlling for intra-individual variability (with random intercepts for participant and random slopes for congruency in the model). This strengthens our suggestion that age-related patterns can not merely be explained by different strategies or intra-individual variability. We refer the reviewer to previous comment 3 for this as well.

6. Additionally, my suggestion to analyze and to illustrate the individual variability is still valid and should be added to the manuscript or at least supplementary material.

Response: We agree with the reviewer that the individual variability is also important to take into account. When using linear mixed models, intra-individual variability is already controlled for, by incorporating random intercepts for participants and slopes for conditions (Baayen et al., 2008). However, we agree that variability in itself can also be an interesting parameter of cognitive performance with age to study as main outcome variable. To explore and illustrate this variability, we have conducted one-way ANOVAs and mixed ANOVAs comparing intra-individual variability between groups and conditions. Variability was indexed here in this study as the standard deviation of reaction times for each task. In addition, for the Go/No-Go and N-Back tasks we also studied and compared standard deviations for commission errors and missed targets, as these are the main indices for inhibition and updating respectively. We have added a detailed description of these additional analyses to the Supplementary materials (under Supplementary Results: Variability as an index of cognitive performance, p.17-21). In general, intra-individual variability increases with older age for all tasks, in line with the literature, except for the commission errors where the opposite pattern was found. For additional analyses on this variability, we refer readers to the dataset on the OSF page (<https://osf.io/vbh58/>), to further scrutinize the data for research questions that go beyond the scope of this manuscript.

In addition, as discussed in the Discussion section of the manuscript, longitudinal data could shed more light on these intra- and individual differences.

Discussion, p. 32-33: “To expand these findings, longitudinal studies are necessary to track trajectories of changes in cognitive functions over time and age, accounting for intra- and inter-individual differences. Moreover, protective and risk factors for cognitive decline such as education, depressive symptoms and social network should be included, as the concept of cognitive reserve already showed the beneficial effect of education and an active lifestyle on cognitive functioning in later life.”

Linking this variability to protective and risk factors and neural measures such as EEG could be an interesting future endeavor. We have added this as a recommendation for future research as well to the Discussion section of the manuscript.

Discussion, p. 33: “Neuroimaging or EEG studies could be interesting in that regard, possibly exposing more subtle neural differences or strategies when exerting cognitive control that differ with older age, going beyond behavioral performance. Our data also allows exploration of other outcome measures for cognitive performance, such as intra-individual variability (Hultsch & MacDonald, 2004; MacDonald et al., 2006). In the Supplementary results we already showed an increase in intra-individual variability in task performance with age, reflected in larger standard deviations of reaction times in older adults. We invite other researchers to further explore the data on the Open Science Framework (<https://osf.io/vbh58/>). Linking this behavioral intra-individual variability to these neural measures and to possible protective and risk factors could further shed light on functional stability and noise with aging in future research (Hultsch & MacDonald, 2004; MacDonald et al., 2006).”

References (additional to those already in the manuscript):

Judd, C. M., Westfall, J., & Kenny, D. A. (2012). Treating stimuli as a random factor in social psychology: a new and comprehensive solution to a pervasive but largely ignored problem. *Journal of personality and social psychology*, 103(1), 54. <https://doi.org/10.1037/a0028347>

Yu, Z., Guindani, M., Grieco, S. F., Chen, L., Holmes, T. C., & Xu, X. (2022). Beyond t test and ANOVA: Applications of mixed-effects models for more rigorous statistical analysis in neuroscience research. *Neuron*, 110(1), 21–35. <https://doi.org/10.1016/j.neuron.2021.10.030>

Meteyard, L., & Davies, R. A. I. (2020). Best practice guidance for linear mixed-effects models in psychological science. *Journal of Memory and Language*, 112, 104092. <https://doi.org/10.1016/j.jml.2020.104092>

Davidson, D. J., & Martin, A. E. (2013). Modeling accuracy as a function of response time with the generalized linear mixed effects model. *Acta psychologica*, 144(1), 83-96. <https://doi.org/10.1016/j.actpsy.2013.04.016>

Bruyer, R., & Brysbaert, M. (2011). Combining Speed and Accuracy in Cognitive Psychology: Is the Inverse Efficiency Score (IES) a Better Dependent Variable than the Mean Reaction Time (RT) and the Percentage Of Errors (PE)? *Psychologica Belgica*, 51(1), 5-13. <https://doi.org/10.5334/pb-51-1-5>

Liesefeld, H. R., & Janczyk, M. (2019). Combining speed and accuracy to control for speed-accuracy trade-offs (?). *Behavior Research Methods*, 51, 40-60. <https://doi.org/10.3758/s13428-018-1076-x>

Baayen, R. H., Davidson, D. J., & Bates, D. M. (2008). Mixed-effects modeling with crossed random effects for subjects and items. *Journal of memory and language*, 59(4), 390-412. <https://doi.org/10.1016/j.jml.2007.12.005>